



Mechanisms of dissolved and labile particulate iron supply to shelf
waters and phytoplankton blooms off South Georgia, Southern Ocean
Christian Schlosser[1,2,*], Katrin Schmidt[3], Alfred Aquilina[1], William B. Homoky[1,4], Maxi
Castrillejo[1,5], Rachel A. Mills[1], Matthew D. Patey[1], Sophie Fielding[3], Angus Atkinson[6], and
Eric P. Achterberg[1,2]
[1] Ocean and Earth Science, National Oceanography Centre Southampton, University of
Southampton, SO14 3ZH Southampton, United Kingdom
[2] GEOMAR Helmholtz Centre for Ocean Research, Wischhofstr. 1-3, 24148 Kiel, Germany
[3] British Antarctic Survey, CB3 0ET Cambridge, United Kingdom
[4] Department of Earth Sciences, University of Oxford, OX1 3AN Oxford, United Kingdom
[5] Institut de Ciència i Tecnologia Ambientals & Departament de Física, Universitat
Autònoma de Barcelona, 08193 Bellaterra, Spain
[6] Plymouth Marine Laboratory, The Hoe, PL1 3DH Plymouth, United Kingdom
* Corresponding author Christian Schlosser (Email: cschlosser@geomar.de,
Phone: 0049 (0) 431 600 1297)





**Abstract (349 words)**
The island of South Georgia is situated in the iron (Fe) depleted Antarctic
Circumpolar Current of the Southern Ocean.  Iron emanating from its shelf system fuels large
phytoplankton blooms downstream of the island, but the actual supply mechanisms are
unclear.  To address this we present the first inventory of Fe, manganese (Mn) and aluminium
(Al) in shelf sediments, pore waters and the water column in the vicinity of South Georgia,
alongside data on zooplankton-mediated Fe cycling processes.  The seafloor sediments were
the main particulate Fe source to shelf bottom waters as indicated by Fe/Mn and Fe/Al ratios
for shelf sediments and suspended particles in the water column.  Less than 1% of the total
particulate Fe pool was leachable surface adsorbed (labile) Fe, and therefore potentially
available to organisms.  Pore waters formed the primary dissolved Fe (DFe) source to shelf
bottom waters supplying $0.1 - 44$ µmol DFe $m^{-2}$ $d^{-1}$.  However, only $0.41 \pm 0.26$ µmol DFe
$m^{-2}$ $d^{-1}$ was transferred to the surface mixed layer by vertical diffusive and advective mixing.
Other trace metal sources to surface waters included glacial flour released by melting glaciers
and zooplankton excretion processes.  On average $6.5 \pm 8.2$ µmol $m^{-2}$ $d^{-1}$ of labile particulate
Fe was supplied to the surface mixed layer via krill faecal pellets, with further DFe released
by krill at around $1.1 \pm 2.2$ µmol $m^{-2}$ $d^{-1}$.  The faecal pellets released by krill constituted of
seafloor derived lithogenic material and settled algae debris, in addition to freshly ingested
suspended phytoplankton specimen. The phytoplankton Fe requirement in the blooms ca.
1,250 km downstream the island of South Georgia was $0.33 \pm 0.11$ µmol $m^{-2}$ $d^{-1}$, with the
DFe supply by horizontal/vertical mixing, deep winter mixing and via aeolian dust estimated
as $\sim 0.12$ µmol $m^{-2}$ $d^{-1}$.  We suggest that additionally required DFe was provided through
recycling of biogenically stored Fe following luxury Fe uptake by phytoplankton on the Fe
rich shelf.  This process would allow Fe to be retained in the surface mixed layer of waters



downstream of South Georgia through continuous recycling and biological uptake, and
facilitate the large scale blooms.



## 1. Introduction

The Southern Ocean is the largest 'High Nitrate Low Chlorophyll' (HNLC) region of
the global ocean (Buesseler et al., 2004), as a consequence of low iron (Fe) supply and
subsequent reduced phytoplankton growth (Buesseler et al., 2004; Tsuda et al., 2009). Iron
can be supplied to surface waters of the Southern Ocean by atmospheric dust inputs (Cassar
et al., 2007; Gao et al., 2001), horizontal/vertical advection and diffusion from Fe enriched
waters (de Jong et al., 2012), resuspension from shelf sediments (Kalnejais et al., 2010;
Marsay et al., 2014), melting of icebergs and glaciers (Raiswell et al., 2008), and
hydrothermal inputs (German et al., 2016). Despite the overall HNLC status of the Southern
Ocean, regions in the wake of islands feature large seasonal phytoplankton blooms; the Fe
sources to these blooms are however poorly constrained (de Jong et al., 2012; Planquette et
al., 2007; Pollard et al., 2009).
Downstream of the island of South Georgia intense, long-lasting phytoplankton
blooms have been observed which extend hundreds of kilometres, and require an enhanced
Fe supply. The blooms peak in austral summer (Borrione et al., 2013), stretch over an area of
ca. 750,000 km$^2$ (Atkinson et al., 2001; Korb et al., 2004), and are responsible for the largest
dissolved inorganic carbon deficit reported within the Antarctic Circumpolar Current (ACC)
(Jones et al., 2015; Jones et al., 2012). As a consequence of the Fe fertilisation the waters in
the vicinity of South Georgia support high biomass with abundant krill and higher predators,
some of which are exploited commercially (Atkinson et al., 2001; Murphy et al., 2007).
South Georgia forms part of the volcanically active Scotia Arc in the Atlantic sector
of the Southern Ocean and is surrounded by a broad 30 to 100 km wide shelf with an average
(albeit highly variable) depth of ca. 200 m (Fig. 1). The island is situated between the
Antarctic Polar Front (PF) and the Southern ACC Front (SACCF), within the general
northeast flow of the ACC (Meredith et al., 2005; Whitehouse et al., 2008). The ACC





surface waters are enriched in nitrate, phosphate and silicic acid, but strongly depleted in
most trace elements, notably Fe and manganese (Mn) (Browning et al., 2014). The large
seasonal phytoplankton blooms downstream of South Georgia are thought to be supplied with
Fe from the island during the passage of ACC waters (Borrione et al., 2013; Nielsdóttir et al.,

2012).

In this study we show the first comprehensive data set of dissolved and (labile)
particulate Fe, Mn, and Al in sediments, pore waters, and the water column overlaying the
shelf and shelf edge regions of South Georgia.  We also include data on the role of krill in
new iron supply and recycling in this region (Schmidt et al., 2011; Schmidt et al., 2016).  We
discuss differences between the various analysed trace metal fractions and quantify dissolved
Fe (DFe) fluxes, such as sedimentary pore water efflux, the supply of sediment derived
particulate Fe to the surface mixed layer, the efflux of Fe from glacial melting and the supply
of Fe by Antarctic krill faecal pellets.  Furthermore, we discuss the productivity of the bloom
region to the north of South Georgia in the relation of the Fe supply rates.

**2. Methods**
**2.1 Cruises and Sampling**
Samples were collected during three research cruises to South Georgia in 2011
(JR247, JC055), and 2013 (JR274).  While cruises JR247 and JR274 aimed to examine the
pelagic shelf ecosystem by collection of predominantly water samples (and zooplankton
during JR247), JC055 explored the composition of sediments on the South Georgia shelf.
Cruise JR247 took place in January 2011 on RRS *James Clark Ross*, and 14 sites on the
northern shelf and shelf edge of South Georgia were visited (stations 1 – 21; Fig. 1).
Suspended particles were collected onto acid cleaned polycarbonate filters (1 μm pore size;
Whatman) using in-situ Stand-Alone Pumping Systems (SAPS; Challenger Oceanic) attached





to a Kevlar wire and deployed at 20 m, 50 m and 150 m depth (Fig. 1, red dots).  The filters
were rinsed with deionized water (Milli-Q; Millipore), stored at -20°C, and shipped frozen to
the National Oceanography Centre Southampton (NOCS).

Subsurface seawater samples were collected by trace metal clean samplers (Ocean

Test Equipment (OTE)) at 9 of the 14 SAPS locations (Fig. 1; black stars). Seawater samples
were filtered using cartridge filter (0.2 μm Sartobran P300; Sartorius) into acid cleaned 125
mL low-density polyethylene (LDPE) bottles (Nalgene).  Unfiltered samples were collected
in 125 mL LDPE bottles for analysis of total dissolvable (TD) trace elements.  Surface waters
from the South Georgia shelf were collected using a tow fish deployed alongside the ship at 3
– 4 m depth.  Samples were filtered in-line using a cartridge filer (0.2 μm Sartobran P300;
Sartorius) into acid washed 125 mL LDPE bottles.  All seawater samples were acidified on-
board with ultra clean $HNO_3$ (15 M UpA grade, Romil) to pH 1.7 (22 μmol $H^+$ $L^{-1}$).

In January and February 2013, RRS *James Clark Ross* cruise JR274 revisited South

Georgia and collected surface seawater samples covering the shelf, shelf-edge, and open
ocean areas around the island.  Dissolved and TD surface seawater samples were collected
using the tow fish and treated similarly to samples from JR247.  For a more detailed
description of all sample-handling procedures, please see Supplementary Text S1.

During the RRS *James Cook* cruise JC055 in February 2011, a megacorer (Bowers

and Connelly type) was used to collect surface sediment and pore water samples on the South
Georgia shelf.  Cores representing the intact sediment – water interface were retrieved from
three sites on the southern shelf, at water depths of ca. 250 m (S1 – S3) (Fig. 1, blue
hexagons).  Pore waters were separated by centrifugation under $N_2$ atmosphere and filtered
using cellulose nitrate syringe filters (0.2 μm pore size; Whatman) (Homoky et al., 2012).
Conjugate sediments were freeze dried on board and stored at room temperature. A more





detailed description of sediment and pore water sample-handling procedures is provided in
Supplementary Text S2.

**2.2 Trace metal analysis in suspended particles**
The labile trace metal fraction of particles was remobilized using a 25% acetic acid
solution (glacial SpA, Romil) following Planquette et al. (2011). This fraction is here after
referred to as the leachable particulate trace metal fraction (LP). The remaining particles
were digested on a hot plate applying a mixture of aqua regia and hydrogen fluoride
(Planquette et al., 2011). This fraction will be referred to as the refractory particulate fraction
(RP). The particulate trace metal fraction (P) is the sum of leachable particulate (LP) and
refractory particulate (RP). All samples were analysed by collision cell inductively coupled
plasma - mass spectrometry (ICP-MS) (ThermoFisher Scientific, XSeriesII).

**2.3 Trace metal analysis of seawater**
The filtered and unfiltered seawater samples were stored for a period of 12 months
prior to analysis. Concentrations of dissolved and total dissolvable Fe, Mn, and Al in
seawater were determined by off-line pre-concentration and isotope dilution / standard
addition ICP-MS (ThermoFisher Scientific Element2 XR) according to Rapp et al. (2017).
For a more detailed description of the method and measured reference materials see
Supplementary Text S1.

**2.4 Trace metal analysis of pore waters and sediments**
Sub-samples of the bulk, homogenized sediments were fully dissolved following an
aqua regia and combined hydrofluoric/perchloric acid digestion method following Homoky et
al. (2011). The acid digests and pore waters were analysed by ICP-optical emission




spectrometry (OES) (Perkin Elmer Optima 4300DV). For a more detailed description of the
method and measured reference materials see Supplementary Text S2.

**3. Results & Discussion**
**3.1 Supply routes of suspended particulate Fe, Mn, and Al**
3.1.1 Two particulate trace metal fractions
Two different particulate fractions were obtained from samples collected during
JR247; the particulate fraction, P, from suspended particles collected using 1 µm pore size
SAPS filters and the leachable particulate fraction from unfiltered seawater samples ($LP_{Un}$) .
$LP_{Un}$ was calculated following Eq. (1):
$$LP_{Un} = \text{total dissolvable (TD)} - \text{dissolved (D; 0.2 µm pore size filters)} \qquad (1)$$
Because of the different sampling approaches and filter cut off sizes (1 µm for SAPS and 0.2
µm for dissolved seawater), concentrations of $LP_{Un}$ and P differed at stations. The
concentrations of Fe, Mn and Al in the $LP_{Un}$ fraction ($LP_{Un}Fe$, $LP_{Un}Mn$, $LP_{Un}Al$) were
slightly lower than the particulate fraction from suspended particles (PFe, PMn, PAl), but
showed similar distribution patterns in the water column (Fig. 2, Table 1 and 2). The $LP_{Un}$
corresponded to ca. $63 \pm 4$ % of the PFe, $83 \pm 11\%$ of the PAl and $100 \pm 10\%$ of the PMn
fractions. The average $LP_{Un}$ trace metal ratios ($LP_{Un}Fe/ LP_{Un}Mn = 33.07 \pm 3.45$ (1 σ) and
$LP_{Un}Fe/ LP_{Un}Al = 0.65 \pm 0.10$ (n=69)), were about half of the elemental ratios of suspended
particles (PFe/PMn = $68.0 \pm 0.6$ and PFe/PAl = $1.251 \pm 0.042$ (n=42) (Fig. 3; Table 1 and

166 2)).

The lower concentrations of Fe and Al in the $LP_{Un}$ compared to the P fractions
suggests that an important fraction of particulate Fe and Al in seawater was not digested
during the acidification procedure at pH 1.7 over 12 months. This refractory particulate
fraction, which represented ~1/3 of particulate Fe and ~ 1/5 of the particulate Al pool, was





likely associated with detrital mineral material that only dissolves during a digestion with
aqua regia and hydrogen fluoride.  This implies that the fraction of $LP_{Un}Fe$ and $LP_{Un}Al$ was
associated to a more labile fraction, such as biogenic and inorganic particles, including
oxyhydroxides (Bonneville et al., 2009; Liu and Millero, 1999), and Fe and Al adsorbed onto
charged surfaces (Schlosser et al., 2011).

Since P and $LP_{Un}$ displayed similar trends with depth and showed a linear

dependencies (Fig. 2 and 3), $LP_{Un}$ was utilized in the following paragraphs as an indicator for
the concentration of particulate trace metals at locations were particulate samples could not
be retrieved by SAPS, e.g. in surface waters and depths greater than 150 m.

3.1.2 Suspended particles in the water column

Concentrations of PFe, PMn and PAl in the water column ranged between 0.87 – 267

nmol $L^{-1}$, 0.01 – 3.85 nmol $L^{-1}$, and 0.60 – 195 nmol $L^{-1}$, respectively (Fig. 2, Table 2).
Concentrations of $LP_{Un}Fe$, $LP_{Un}Mn$ and $LP_{Un}Al$ ranged between 1 – 118 nmol $L^{-1}$, 0.01 – 100
nmol $L^{-1}$, and 1 – 141 nmol $L^{-1}$, respectively (Fig. 2, Table 1).  Below the isopycnal density
layer 27.05 kg $m^{-3}$ (at ca. 50 – 70 m depth), P and $LP_{Un}$ increased with depth and showed a
maximum near the seafloor of e.g. 207 nmol $L^{-1}$ for PFe and 112 nmol $L^{-1}$ for $LP_{Un}Fe$ (#17,
Table 2).  Most stations on the shelf (bottom depth ≤ 260 m; #9/10, #13, #14, #17, and #21)
showed seafloor maxima, in agreement with other shelf studies.  For example, Milne et al.
(2017) reported concentrations of up to 140 nmol $L^{-1}$ for PFe and 800 nmol $L^{-1}$ for PAl in
bottom waters on the west African shelf, and Chase et al. (2005) showed bottom water
maxima of up to 400 nmol $L^{-1}$ for $LP_{Un}Fe$ off the Oregon coast.

Strong linear relationships between elements were observed for suspended particles

(SAPS) obtained from above and below the isopycnal, with elemental ratios of PFe/PMn =
68.0 ± 0.6 and PFe/PAl = 1.25 ± 0.04 (n=42) (Fig. 3, Table 2). The elemental ratio were



comparable to those reported for the earth crust (Fe/Mn = 60.0 ± 0.2 (Wedepohl, 1995)),
indicating that suspended particles had a lithogenic source.

The elemental ratios of suspended particles were higher than those for sediments

(mean sediment surface layer of S1, S2, S3; SFe/SMn = 51.5 ± 2.4 and SFe/SAl = 0.34 ± 0.02
(Fig. 4, Table 3)).  The Fe/Mn ratios among different phytoplankton species show strong
variations but are typically much lower (Fe/Mn ~ 1.7 (Ho et al., 2003)) with lower Fe
concentrations than terrestrial/sediment particles (cellular concentration of phytoplankton ~
0.7 mmol kg$^{-1}$ (Ho et al., 2003); upper crust ~ 550 mmol kg$^{-1}$ (Wedepohl, 1995)).  A
prevalence of biogenic particles in the suspended particle pool would be expected to lower
the PFe/PMn ratio in our fully digested samples to values less than 51.5.

It is likely that enhanced scavenging of DFe onto lithogenic/sediment particles

increased the Fe to Mn (and Fe to Al) ratio of suspended particles (PFe/PMn = 68.0)
compared to sediment particles (SFe/SMn = 51.5).  At seawater pH 8, dissolved Fe(III) is
rapidly hydrolysed to soluble Fe(III)(OH)$_3$ (< 0.02 μm) which readily accumulates as
nanometer sized colloids (0.02 – 0.2 μm) (Liu and Millero, 2002).  It has been shown that
both soluble and colloidal Fe are attracted by charged surfaces, a process that lowers the
overall amount of DFe and simultaneously increases the amount of particulate Fe in seawater
over time (Schlosser et al., 2011).

A range of mechanisms delivers suspended particles to the surface waters.  These

transport mechanisms will be discussed in the following section.

3.1.3 Glacial outflow and zooplankton activity

Whilst most stations on the shelf showed bottom water maxima of suspended

particles, at three sampling sites located on the shelf (#18) and shelf edge (#15/16 and
#19/20), the particulate trace metal concentrations featured maxima in the top 100 m of the



water column (Fig. 2 and 5). At station #19/20, ca. 100 km away from the coast with a water
depth of 1741 m, the PFe concentration at 20 m depth was 97 nmol L$^{-1}$, similar to LP$_{Un}$Fe
(Fig. 5).  The elemental ratio PFe/PMn of these samples (e.g. 64.2 for station #19/20, 20 m
depth) were close to the average ratio (PFe/PMn = 68.0), indicating that lithogenic particles
dominated the suspended particulate pool in these surface waters.

The surface water maxima could have two supply routes: 1) lateral transport of waters

containing lithogenic particles from shallow island shelf sediments, and 2) transport of glacial
particles following melt processes.  The reduced salinities (~32.5) recorded in surface waters
in Cumberland Bay and ~50 km offshore of South Georgia (~33.6) (Fig. 6(c)) provide an
indication of glacial outflow, melting of icebergs and run-off of melt water streams.
Enhanced LP$_{Un}$Fe concentrations of up to 22 μmol L$^{-1}$ in low salinity surface waters of
Cumberland Bay (Fig. 6(a)), are indicative of a meltwater source.  The LP$_{Un}$Fe concentration
decreased strongly with increasing distance from the coast, and exhibited an abrupt reduction
to 1 – 5 nmol Fe L$^{-1}$ at the shelf edge ca. 100 km offshore. A similar distribution pattern was
observed for LP$_{Un}$Mn (Fig. 6(d)) and LP$_{Un}$Al (not shown), for cruises JR247 and JR274.
Glacial melt has been reported as an important source of particulate material in the vicinity of
the Antarctic Peninsula (de Jong et al., 2012). For example, Gerringa et al. (2012)
documented elevated total dissolvable Fe concentration of up to 106 nmol L$^{-1}$ near the Pine
Island Glacier in the Amundsen Sea, and Raiswell et al. (2008) estimated that per year 1.6
Gmol nanoparticulate Fe, associated to terrigenous particles, are delivered to the Southern
Ocean by melting ice.

Locally elevated particulate metal concentrations in surface waters may also be

related to production of faecal pellets by swarms of Antarctic krill (*Euphausia superba*)
(Schmidt et al., 2016).  High abundances of Antarctic krill estimated from acoustic
backscattering (Fielding et al., 2014) and large numbers of faecal pellets were observed on





the SAPS filters during cruise JR247. The stomach content of Antarctic krill contained up to
80% sediment particles by volume, an observation that was attributed to feeding by these
organisms on deep ocean sediments (Schmidt et al., 2011) and glacial flour (Schmidt et al.,
2016). Krill thus take up lithogenic particles incidentally during filter feeding on their
phytoplankton food and transfer and suspend them in the surface ocean following their
ascend through excretion of faecal pellets (Schmidt et al., 2016). The trace metal contents of
krill faecal pellets collected during on-board incubation experiments during JR247 ranged
between 0.88 – 67.14 µg Fe mg$^{-1}$ dry weight (n = 27) (Table 4). The molar ratios PFe/PMn =
70.5 ± 8.21 and PFe/PAl = 0.48 ± 0.07 were similar to those for LP$_{Un}$ in bottom waters and P
in suspended particles (Table 1, 2 and 4), indicating that krill faecal pellets predominately
contained sediment and/or glacial flour particles.

**3.2 Supply routes of dissolved Fe, Mn, and Al**
Concentrations of DFe, DMn, and DAl in the water column showed strong variations
and ranged between ca. 0.1 – 7.7 nmol L$^{-1}$, 0.3 – 2.1 nmol L$^{-1}$ and 0.1 – 18.4 nmol L$^{-1}$,
respectively (Fig. 2, 5 and 7). Dissolved Fe and Mn in the surface waters ranged between 0.1
– 25.9 nmol L$^{-1}$ and 0.1 – 19.6 nmol L$^{-1}$, respectively, and were highest in Cumberland Bay,
and lowest beyond the shelf break (Fig. 6). Dissolved Fe concentrations from this study are
in agreement with reported DFe near the Antarctic Peninsula (0.6 – 14.6 nmol L$^{-1}$ (de Jong et
al., 2012)) and Crozet Islands (0.1 – 2.5 nmol L$^{-1}$ (Planquette et al., 2007)). Sources and
sinks of dissolved trace metals, and their distribution in the water column are discussed in the
following sections.

3.2.1 Supply from sediment pore waters



Elevated pore water concentrations of Fe and Mn ($Fe_{PW}$ and $Mn_{PW}$) were observed in
sediments from shelf sites at water depths of around 250 m, and ranged between 0.5 – 110
µmol kg$^{-1}$ for Fe and 0.1 – 2 µmol kg$^{-1}$ for Mn (Fig. 7 and Table S2). The down-core
distributions of $Fe_{PW}$ and $Mn_{PW}$ were consistent with microbial dissimilatory Mn and Fe
reduction during organic matter oxidation (Canfield and Thamdrup, 2009), and thus
concentrations were elevated at defined depth horizons controlled by their redox potential
(Eh) (Bonneville et al., 2009; Raiswell and Canfield, 2012). The $Fe_{PW}$ and $Mn_{PW}$
concentrations near the sediment-seawater interface were used to calculate fluxes of Fe and
Mn to bottom waters following diffusion of reduced Fe and Mn species across an oxygenated
layer in surface sediments. These calculations were performed following Boudreau and Scott
(1978) and Homoky et al. (2012), and are described in detailed in the Supplementary material
(Text S3 and Table S1). We are aware that our calculated fluxes represent minimum
estimates of pore water efflux, which under natural conditions is supplemented by advection
due to bioirrigation, bioturbation, and bottom water currents (Homoky et al., 2016).
We calculated substantial benthic fluxes from sediment pore waters to bottom waters
for $Fe_{PW}$ of <0.1 to 44.4 µmol m$^{-2}$ d$^{-1}$ and $Mn_{PW}$ of 0.6 to 4.1 µmol m$^{-2}$ d$^{-1}$. The upper flux
values for Fe are comparable to those reported for dysoxic and river-dominated continental
margins (3.5 – 55 µmol m$^{-2}$ d$^{-1}$ (Homoky et al., 2012)), seasonal maxima of temperate and
oxic shelf seas (23 – 31 µmol m$^{-2}$ d$^{-1}$ (Klar et al., 2017)), and shelf sediments off the Antarctic
Peninsula (1.3 – 15.5 µmol m$^{-2}$ d$^{-1}$ (de Jong et al., 2012)). The Mn fluxes were relatively low
for shelf environments, with for example fluxes of 70 – 4450 µmol m$^{-2}$ d$^{-1}$ reported for Baltic
and Black Sea sediments (Pakhomova et al., 2007)). The substantial Fe pore water fluxes
from the South Georgia shelf sediments, which extend over an area of ca. 40,000 km$^2$,
indicate that these may serve as an important year-round source to overlying waters, totalling
4 to 1,728 kmol DFe d$^{-1}$ and 25 to 164 kmol DMn d$^{-1}$.



295   Benthic release of trace metal enriched pore waters shaped the distributions of

296 dissolved trace metals in bottom waters on the shelf. Concentrations of DFe, DMn, and DAl

297 were enhanced at isopycnals > 27.05 kg m$^{-3}$ (e.g. DFe up to 7.70 nmol L$^{-1}$ at station #21,

298 Table 1) compared to surface waters (e.g. DFe as low as 0.30 nmol L$^{-1}$ at #13, Table 1; Fig. 2

299 and 7). Trace metal enriched bottom waters were also observed at site #13, #14, #17 and #18

300 (Fig. 2). The molar DFe/DMn ratios in oxygenated bottom waters varied between 1.1 – 3.5

301 and were thus similar to pore waters (0 – 1 cm depth) near the sediment-seawater interface

302 (Fe$_{PW}$/Mn$_{PW}$ = 2.2 ± 1.0; Fig. 7). The similar trace metal ratios suggests that Fe and Mn in

303 enriched pore waters crossed the sediment-bottom water interface and accumulated in shelf

304 bottom waters.

305   To determine the vertical DFe fluxes from near bottom to surface waters we

306 employed a method outlined by de Jong et al. (2012), and calculated both the advective and

307 diffusive flux terms. Applying literature values for vertical diffusivity ($K_Z$ = 1 x 10$^{-4}$ m$^2$ s$^{-1}$

308 (Charette et al., 2007)) and upwelling velocity ($w$ = 1.1 x 10$^{-6}$ m s$^{-1}$ (de Jong et al., 2012))

309 yielded an average vertical DFe flux on the shelf of 0.41 ± 0.26 μmol m$^{-2}$ d$^{-1}$ from subsurface

310 waters into the surface mixed layer (Supplementary Text S4). The surface mixed layer depth

311 was determined by a density criteria (~0.03 kg m-3 (de Boyer Montégut et al., 2004)) and

312 was located ca. 50 m depth. About 38% of the DFe flux was related to Ekman upwelling

313 (advective term) and 62% to the diffusive flux. This vertical flux is at the lower end of the

314 calculated benthic flux from this study (Fe$_{PW}$ < 0.1 to 44.4 μmol m$^{-2}$ d$^{-1}$), and agrees with

315 values reported for other Southern Ocean shelf regions near the Antarctic Peninsula (within

316 20 – 70 km from the coast: ~ 2.7 ± 3.4 μmol m$^{-2}$ d$^{-1}$ (de Jong et al., 2012)) and the Crozet

317 Islands (only diffusive flux of 0.06 μmol m$^{-2}$ d$^{-1}$ (Planquette et al., 2007)).


319 3.2.2 DFe supply from suspended particles



The analytical protocol for analysis of the particulate material obtained using SAPS
yielded refractory and leachable fractions (RP and LP, respectively).  The RP fraction of the
suspended matter is considered to include silicates and aged oxide minerals, and the LP
fraction represents predominantly oxyhydroxides, biogenic material and loosely bound
surface associated elements which are readily remobilized using leaching procedures (Berger
et al., 2008).
Concentrations of LPFe, LPMn and LPAl in the water column showed strong
variations, ranging from a few picomoles to several nanomoles $L^{-1}$ (Table 2).  On average,
LPFe and LPAl concentrations at 150 m depth (~ 1.3 nmol LPFe $L^{-1}$ and ~0.95 nmol LPAl $L^{-1}$
) were significantly higher than at 20 and 50 m (LPFe = 0.3 nmol $L^{-1}$ (student t-test:
t(0.95;28) = 1.725 (1.703)); LPAl = 0.43 nmol $L^{-1}$ (student t-test: t(0.90;28) = 1.383
(1.313))).  The LPMn concentrations did not change strongly and remained near constant
throughout the top 150 m (LPMn = 8.9 pmol $L^{-1}$ (student t-test: (0.65;28) = 0.400 (0.390))).
The average contribution of LP to the particulate pool was low; 0.83 ± 1.13% for Fe, 2.55 ±
1.58% for Mn and 2.42 ± 1.32% for Al (Table 2).  A study conducted in the North Pacific
near the Columbia River outflow reported considerably higher LP fractions (e.g. 6.6±3.0% of
Fe, 78.7±14.0% of Mn, 6.3±2.0% of Al (Berger et al., 2008)), which was attributed to
enhanced biogenic particle levels in the low salinity waters of the river (Berger et al., 2008).
In contrast, results from our study showed that particulate trace metals mainly had a
refractory component (RP), indicating that Fe, Mn, and Al was mainly incorporated in
lithogenic material.
A weak linear relationship between RP and LP was observed for Fe ($R^2$ = 0.57), Mn
($R^2$ = 0.64) and Al ($R^2$ = 0.63) (Supplementary Fig. S1), indicating that the LP fraction
included mainly Fe, Mn and Al that was scavenged onto lithogenic particle surfaces and not
much LPFe was incorporated in biogenic particles. The scavenging of dissolved trace metals




by charged particle surfaces is established (Homoky et al., 2012; Koschinsky et al., 2003),
but how well Fe and other trace metals can be remobilized from marine particle surfaces and
which process may modify their availability over time is not yet constrained.
Freshly produced inorganic Fe(III) oxyhydroxide precipitates in seawater are subject
to chemical and structural conversions that lead to less soluble particles with time (Yoshida et
al., 2006). Scavenged Fe is however also reported to buffer DFe concentrations in the water
column of the tropical Atlantic (Milne et al., 2017). Furthermore, recent work has indicated
that zooplankton grazing and the production of faecal pellets remobilizes DFe from lithogenic
and biogenic particles (Giering et al., 2012; Riley et al., 2012; Schmidt et al., 2016).

### 355 3.2.3 DFe supply from Antarctic krill

Elevated dissolved trace metal concentrations in the top 200 m of the water column
coincided with elevated particulate concentrations at stations #11/12, #15/16, #18, and #19/20
(Fig. 2, 5, and 7). The SAPS filters from these stations contained a high load of krill faecal
pellets. To elucidate the relationship between dissolved trace metal concentrations and
abundance of Antarctic krill and krill faecal pellets, krill were caught and incubated on-board
the vessel as described in Schmidt et al. (2016).
Krill excretion rates of DFe were variable, relating positively to recent ingestion of
diatoms. However, on average krill released ~2.0 ± 1.9 nmol DFe individual$^{-1}$ d$^{-1}$ (Schmidt et
al., 2016). By applying an average Antarctic krill abundance of 465 ± 588 individuals m$^{-2}$,
estimated from acoustic backscattering (Fielding et al., 2014), krill excreted 1.1 ± 2.2 µmol
DFe m$^{-2}$ d$^{-1}$ into the top 300 m of the water column (Schmidt et al., 2016). In addition, krill
produced ca. 1.8 ± 1.6 mg of faecal pellets per individual per day. Particle leaches performed
on those faecal pellet samples with 25% acetic acid showed that on average 2.5 ± 2.1% of the
total Fe in these pellets could be remobilised (Table 4), which would equate to a production



of 14 ± 24 nmol LPFe ind$^{-1}$ d$^{-1}$.  By multiplying the mean LPFe by the ambient krill density
used above, we calculate a LPFe flux of 6.5 ± 8.2 µmol m$^{-2}$ d$^{-1}$ from the faecal pellets to the
water column.

Since krill are mobile animals, questions remain over where the major part of the

LPFe flux occurs, and what the fate of this Fe source is.  Highest krill abundances were
recorded generally (but not exclusively) in the top 100 m layer (Fielding et al., 2014), and
hence a large proportion of this LPFe flux from krill is likely to occur in the upper waters.
Notwithstanding our current uncertainties over the depths of origin and fate, the LPFe flux
from krill fecal pellets and the release of DFe were on average an order of magnitude higher
than the vertical diffusive and advective DFe flux from below, illustrating the potential
importance of zooplankton-mediated-Fe-cycling, in agreement with previous studies
(Hutchins and Bruland, 1994; Sato et al., 2007).

**3.3 Off-shore transport of trace metal enriched water masses**

Along the NE – SW transect (Fig. 1; #11/12 via # 13 to #14), lateral water mass

transport carried suspended particles offshore.  Indeed, elevated concentrations of the P and
LP$_{Un}$ metal fractions were observed in subsurface waters that had been in recent contact with
the shelf.  These metal enriched waters, detected at the eastern shelf edge station #11/12
between 200 and 400 m water depth (Fig. 1 and 4), exhibited similar temperature and salinity
signatures to shelf bottom waters.  Furthermore, the elemental ratios of the LP$_{Un}$ fraction in
these waters were similar to the particles in the surface sediments (S1, S2, and S3) and the
resuspended particles in the bottom boundary layer (#13 and #14) on the shallow shelf (Fig.
4).  A similar distribution was also found for the P fractions, but limited to station #13 and
#14, as SAPS were not deployed below 150 m at the shelf edge location #11/12.





The LP$_{Un}$Fe concentration decreased exponentially with distance from the island to
the offshore: ($P_{Un}Fe = 267.7 * e^{-0.047*d}$, $R^2 = 0.999$), from station #14 in 200 m depth
($P_{Un}Fe = 82.26$ nmol L$^{-1}$) to #13 in 100 m depth ($P_{Un}Fe = 34.06$ nmol L$^{-1}$) to #11/12 between
200 and 400 m depth ($P_{Un}Fe = 10.18$ nmol L$^{-1}$) (Table 1). The variable $d$ represents the
distance to the coast in kilometres. A similar exponential decrease was observed for the
SAPS data: ($PFe = 125.02 * e^{-0.056*d}$, $R^2 = 1$), from station #14 ($PFe = 31.12$ nmol L$^{-1}$) to
#13 ($PFe = 10.23$ nmol L$^{-1}$). The decrease of P and LP$_{Un}$ with increasing distance to the coast
is in agreement with previous observations in the Western Subarctic Pacific (Lam and
Bishop, 2008), which reported elevated LPFe concentrations in the range between 0.6 and 3
nmol L$^{-1}$ in subsurface waters between 100 and 200 m depth along the Kamchatka shelf and
related this observation to offshore water mass transport.
Consistent with the observed P and LP$_{Un}$ distributions, elevated dissolved metal
concentrations at depths between 200 and 400 m at station #11/12 indicated that trace metal
enriched shelf bottom waters were transported offshore (Fig. 7). For horizontal flux
calculations we used the entire DFe data set for water depth between 100 and 400 m.
However, average DFe concentrations in this depth range were highly variable and did not
follow an exponential or power law function with distance from the coast (Fig. S3), which is
necessary to determine scale length and horizontal diffusivity ($K_h$) (de Jong et al., 2012). As
a result, horizontal flux calculations from the data could not be executed. Even though flux
estimate for this study are not available, the overall distribution of DFe in surface waters
might help to determine the horizontal transport of DFe across the shelf break.
The distribution of dissolved trace metals in surface waters indicated that a limited
transfer of DFe beyond the shelf break into the bloom region. Surface samples showed that
DFe concentrations were strongly enriched in surface waters on the shelf (0.3 – 25.9 nmol L$^{-1}$
$^1$, Fig. 6(b)), while DFe concentrations beyond the shelf break decreased abruptly to



concentrations below 0.2 nmol L$^{-1}$ (Fig. 6(b)). This indicates that DFe was quickly removed
from ACC surface waters following passage of the island. However, previous studies in the
region indicated DFe transfer beyond the shelf break of South Georgia (Borrione et al., 2013;
Nielsdóttir et al., 2012). Nielsdóttir et al. (2012) reported surface waters downstream the
island shelf with up to 2 nmol L$^{-1}$ DFe, with seasonal variations and highest concentrations
during austral summer in January/February 2008. Dissolved Fe data from JR247 (2011) and
JR274 (2012) were also obtained during the summer season, but indicated rapid reduction in
DFe concentrations through mixing, biological uptake and/or particle scavenging.

**3.4 Iron budget in the bloom region**
Large seasonal phytoplankton blooms downstream of South Georgia recorded by
earth observing satellites are initiated by Fe supplied from the South Georgia island/shelf
system during the passage of ACC waters (Fig. 1) (Borrione et al., 2013; Nielsdóttir et al.,
2012). Based on our study, the main DFe sources during this passage of the ACC were
benthic release and vertical mixing, release of DFe from krill and krill faecal pellets, and
supply of particles from run-off and glacial meltwater. In the following sections we will
discuss the strength of each DFe source in the bloom region ca. 1,250 km downstream of the
island and estimate how much DFe is required to stimulate the elevated primary productivity
in that region.

3.4.1 Phytoplankton Fe requirements in the blooming region
The surface ocean in the vicinity of South Georgia during the austral summer features
strongly elevated biomass production (Gilpin et al., 2002) and represents the largest known
CO$_2$ sink in the ACC (12.9 mmol C m$^{-2}$ d$^{-1}$ (Jones et al., 2012)). The Fe requirements of the
phytoplankton community in the austral summer within the bloom ca. 1,250 km downstream



the island were estimated by combining satellite-derived marine primary productivity data
($62 \pm 21$ mmol C m$^{-2}$ d$^{-1}$ (Ma et al., 2014)) with an average intracellular Fe:C ratio obtained
from five Southern Ocean diatom species ($5.23 \pm 2.84$ µmol Fe mol$^{-1}$ C (Strzepek et al.,
2011)). This approach yielded an approximate Fe requirement of $0.33 \pm 0.11$ µmol Fe m$^{-2}$ d$^{-1}$
for the phytoplankton community (Fig. 8). For a more detailed description of the applied
values and calculations see Supplementary Text S4.

3.4.2 Horizontal and vertical mixing

De Jong et al. (2012) reported that horizontal and vertical advective, diffusive

(diapycnal) and deep winter mixing downstream ($1,250 - 1,570$ km) of the Antarctic
Peninsula (between 51°S and 59°S) supplied DFe to the surface waters in quantities that
exceeded the DFe requirement of primary producer ($0.13 \pm 0.04$ µmol DFe m$^{-2}$ d$^{-1}$) during
austral summer. In their study region, de Jong et al. (2012) determined that ca. $0.30 \pm 0.22$
µmol DFe m$^{-2}$ d$^{-1}$ were supplied by horizontal and vertical fluxes, of which 91% of the
vertical flux were attributed to Ekman upwelling (advective term), and 43% of the entire DFe
flux was supplied by deep winter mixing. Tagliabu et al. (2014) reported similar model
estimates for the region that is located south of the Polar Front and characterized by strong
Ekman upwelling and winter entrainment.

For the bloom region downstream of South Georgia model calculations by Tagliabue

et al. (2014) indicated that less than 0.0003 µmol DFe m$^{-2}$ d$^{-1}$ were supplied by diapycnal
mixing, and ca. -0.0028 µmol DFe m$^{-2}$ d$^{-1}$ were removed by Ekman downwelling. For the
vertical flux component, this yields an overall loss of DFe of -0.0025 µmol DFe m$^{-2}$ d$^{-1}$ in the
blooming region north of South Georgia (Fig. 8).

Because our horizontal flux calculations were invalid, we applied the horizontal flux

estimates from de Jong et al. (2012) for our own Fe budget. For a region ca. 1,250 km





downstream of a source, calculations according to de Jong et al. (2012) indicate that ca. 0.11
± 0.03 µmol DFe m$^{-2}$ d$^{-1}$ are supplied to the bloom region by horizontal advection and
diffusion (Fig. 8).
3.4.3 Deep winter mixing

The entrainment of new DFe during winter represents an important Fe source to

surface waters in the Southern Ocean (de Jong et al., 2012; Tagliabue et al., 2014). Elevated
DFe concentrations in subsurface waters support primary production in the austral spring
following entrainment by deep winter mixing. Model estimates showed that DFe supplied by
winter mixing together with diapycnal mixing matches the Fe requirements at most low
productivity sites in the Southern Ocean. However, deep winter mixing at the high
productive sites north of South Georgia supplies only ca. 0.011 µmol m$^{-2}$ d$^{-1}$ (Tagliabue et al.,
2014) (Fig. 8). Later in the season primary productivity in surface waters is considered to rely
strongly on Fe derived from recycling of biogenic material (Boyd et al., 2015).

3.4.4 Dust deposition

Dissolved Fe supplied by the deposition of aeolian dust is considered to be an

important source to the Southern Ocean (Conway et al., 2015; Gabric et al., 2010; Gassó and
Stein, 2007). Aeolian flux estimates, applied by Borrione et al. (2013) for their South
Georgia regional model, suggested that up to 8 µmol Fe m$^{-2}$ d$^{-1}$ are delivered to the bloom
regions downstream of South Georgia by dry and wet deposition. However, reliable dry and
wet deposition estimates for the Southern Ocean are limited. Data from the South Atlantic
along 40°S, ca. 600 nm north of South Georgia, showed that rather low levels of DFe (~
0.002 µmol m$^{-2}$ d$^{-1}$) are supplied by dry deposition (Chance et al., 2015). On the other hand,
ca. 1.0 ± 1.2 µmol DFe m$^{-2}$ d$^{-1}$ are delivered sporadically to the 40°S area by wet deposition
(Chance et al., 2015). However, even when assuming that similar wet deposition fluxes



occur north of South Georgia, fertilization with DFe is temporally and spatially limited.
Furthermore, it is very unlikely that such sporadic events could cause long-lasting and far
extending phytoplankton blooms strictly constrained between the PF and the SACCF.
3.4.5 Luxury Fe uptake on the shelf

Vertical/horizontal mixing, deep winter entrainment and dust deposition together

supply significantly less DFe (< 0.12 µmol Fe m$^{-2}$ d$^{-1}$) into the bloom region than the
phytoplankton community requires (~ 0.33 µmol Fe m$^{-2}$ d$^{-1}$) (Fig. 8).  The missing supply of
ca. 0.21 µmol DFe m$^{-2}$ d$^{-1}$ is likely laterally supplied to the bloom region through advecting
phytoplankton cells that are enriched in labile Fe.  It has been demonstrated that Fe-rich
biogenic particles can be created by luxury iron uptake of diatoms (Iwade et al., 2006;
Marchetti et al., 2009).  Using bottle incubation experiments, Iwade et al. (2006) showed that
under high Fe conditions the coastal diatom *Chaetoceros sociale* stores more intracellular Fe
than needed for the production of essential enzymes and proteins.  We therefore suggest that
phytoplankton cells that grew under excess nutrient supply on the South Georgia shelf stored
more Fe than needed for their metabolic processes and that via remineralisation this iron is
remobilised in surface waters and made available for phytoplankton uptake.

High recycling efficiencies, described by the f*e* ratio (Boyd et al., 2005), are required

to maintain the cycle of remineralisation and uptake in the euphotic zone.  This counteracts
the loss of particulate Fe by vertical export.  Boyd et al. (2015) reported the highest recycling
efficiencies of ca. 90% for cold and low-DFe waters such as downstream of South Georgia.
Further, these workers showed that the degree of recycling is controlled by the abundance of
bacteria with a high Fe quota, such as prokaryotic cyanobacteria, and particularly by grazing
zooplankton. The waters off South Georgia feature among the highest biomasses worldwide
of metazoan grazers; (Atkinson et al., 2001). These large grazers, chiefly copepods and krill,



are able to ingest large, Fe rich diatoms (Atkinson, 1994; Hamm et al., 2003), thereby
disintegrating cell membranes and releasing trace metals.

In recent years it has become apparent that the recycling of biogenic particles in the

euphotic zone is a critical mechanism that maintains primary production, especially when the
dissolved nutrient pools become exhausted (Boyd et al., 2015; Tagliabue et al., 2014).
However, uncertainties remain to the degree to which Fe is lost during each cycle of uptake
and remineralisation. Thus more research is needed, especially field work that encompasses
the community structures (bacteria, phytoplankton, zooplankton, and higher predators
(Ratnarajah et al., 2017; Wing et al., 2014)), the degree of recycling for macro- and micro-
nutrients in the euphotic zone, and loss of Fe through vertical export.

**4. Conclusions**

Shelf sediment-derived Fe and Fe released from Antarctic krill controls the DFe

distribution in the shelf waters around South Georgia. Nevertheless, DFe enriched in shelf
waters are not effectively advected to the phytoplankton bloom region downstream of the
island.  Together with other Fe supplies, such as aeolian dust, deep winter mixing and
diapycnal mixing, the horizontal advection contributes insufficiently to the Fe requirements
of the bloom.

The majority of the Fe appears to be derived from remineralisation of Fe enriched

phytoplankton cells/biogenic particles that are transported with the water masses into the
bloom region.

In the 1920s the scientists of the Discovery Investigations speculated that micro-

nutrients were responsible for the high productivity near South Georgia (Hardy and Gunther,
1935).  Identifying the cause of South Georgia productivity is important because the
conditions around this island are changing rapidly. Summer water temperatures have





increased by more than 0.9°C since the 1920s (Whitehouse et al., 1996). Glaciers are in
retreat (Cook et al., 2010; Hodgson et al., 2014) and populations of larger zooplankton
(Atkinson et al., 2004) and higher predators (Forcada and Hoffman, 2014; Murphy et al.,
2007) have diminished substantially. Each of the potential nutrient sources may change
differently, for example glacial outflow will change non-linearly in a changing climate and
the increase in positive Southern Annular Mode anomalies in recent decades (Gillett and
Fyfe, 2013) indicates increasing westerlies that may transport more Aeolian dust to the
Southern Ocean. While we highlight the importance of grazers and the cycling of various
particulate Fe phases in the Fe-fertilisation of the South Georgia bloom, more work is needed
to clarify the transport mechanisms of dissolved and particulate Fe.

**Author contribution**
CS, KS, EA and AA designed the experiments for JC247. CS, MP and AAt carried the
experiment out during JC247 and CS and MC analysed the trace metal samples at NOCS. EA
carried the experiment out during JC274. Samples from JC274 were analysed by CS and MC.
AAq, WH and RM designed the experiments for JR55 and AAq analysed the samples. CS
prepared the manuscript with contributions from all co-authors.
**Acknowledgements**
We would like to thank the officers and crew of RRS *James Clark Ross* for assistance
with the pelagic sampling and those of RRS *James Cook* for the benthic coring. This work
forms part of the NERC-AFI grant AFI9/07 to AA and EA (NE/F01547X/1). RAM was
funded by NERC grants NE/01249X/1 and NE/H004394/1. WBH was supported by NERC
fellowship NE/K009532/1



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





**Table 1:** Fe, Mn, and Al concentrations determined for the dissolved (D) (0.2 μm) and the
leachable particulate fraction (LP$_{UN}$) (total dissolvable – dissolved) of unfiltered seawater
samples collected during JR247. Additional information covers sampling date, station ID,
event number and latitude and longitude.

| Date | Station ID | Depth | Leach. Part. (nmol L⁻¹) | | | Dissolved (nmol L⁻¹) | | |
|---|---|---|---|---|---|---|---|---|
| | Lat. & Lon. | (m) | LP$_{Un}$Fe | LP$_{Un}$Mn | LP$_{Un}$Al | DFe | DMn | DAl |
| 04/01/2011 | #9/10 (E95 & E97) | 20 | 20.36 | 0.95 | 46.41 | 5.71 | 1.83 | 1.11 |
| | | 50 | 15.18 | 0.42 | 40.86 | 3.19 | 1.88 | 2.27 |
| | 54.26°S, 35.35°W | 100 | 9.86 | 0.23 | 20.43 | 1.55 | 0.92 | 2.07 |
| | | 130 | 23.33 | 0.73 | 48.91 | 2.82 | 0.87 | 2.68 |
| | | 150 | 23.71 | 0.43 | 46.95 | 2.35 | 1.03 | 0.12 |
| | | 200 | 27.37 | 0.62 | 54.41 | 2.70 | 0.89 | 2.37 |
| 05/01/2011 | #11/12 (E98 & E101) | 20 | 4.05 | 0.38 | 6.68 | 2.19 | 0.41 | 3.57 |
| | | 35 | 1.52 | 0.39 | 7.28 | 0.41 | 0.37 | - |
| | 54.62°S, 34.81°W | 50 | 9.30 | 0.60 | 22.20 | 7.18 | 0.64 | 13.31 |
| | | 75 | 1.28 | 0.31 | 7.85 | 0.77 | 0.35 | 4.56 |
| | | 100 | 2.02 | 0.32 | 3.34 | 1.09 | 0.35 | 1.47 |
| | | 150 | 1.55 | 0.38 | 3.18 | 1.10 | 0.45 | - |
| | | 200 | 13.10 | 1.31 | 23.81 | 1.26 | 1.17 | 3.07 |
| | | 300 | 8.62 | 0.70 | 23.25 | 1.06 | 0.55 | - |
| | | 400 | 8.81 | 0.54 | 16.54 | 2.05 | 0.46 | 2.69 |
| | | 500 | 4.51 | 0.41 | 11.41 | 0.72 | 0.38 | 0.76 |
| | | 600 | 2.75 | 0.37 | 10.32 | 0.96 | 0.36 | 0.77 |
| | | 700 | 4.81 | 0.41 | 16.85 | 0.82 | 0.35 | - |
| 06/01/2011 | #13 (E105) | 20 | 3.46 | 0.62 | 14.68 | 0.28 | 0.57 | 4.53 |
| | | 35 | 1.00 | 0.33 | 7.17 | 0.10 | 0.28 | 2.64 |
| | 54.53°S, 35.27°W | 50 | 7.09 | 0.71 | 22.62 | 1.26 | 0.57 | 5.77 |
| | | 75 | 25.03 | 1.09 | 61.94 | 1.23 | 0.64 | 5.86 |
| | | 100 | 34.06 | 1.30 | 87.43 | 0.82 | 0.74 | 4.08 |
| 07/01/2011 | #14 (E113) | 20 | 4.00 | 0.89 | 7.87 | 0.64 | 0.85 | 2.57 |
| | | 50 | 2.23 | 0.31 | 7.64 | 0.27 | 0.32 | 1.80 |
| | 54.56°S, 35.59°W | 75 | 2.30 | 0.43 | 3.58 | 0.62 | 0.46 | 2.42 |
| | | 100 | 2.26 | 0.44 | 3.34 | 0.35 | 0.46 | 0.46 |
| | | 150 | 23.50 | 0.94 | 33.35 | 0.70 | 0.62 | 0.23 |
| | | 200 | 82.26 | 2.12 | 103.11 | 2.69 | 0.77 | 2.31 |
| 08/01/2011 | #15/16 (E119 & E129) | 20 | 17.66 | 0.46 | 26.66 | 0.99 | 1.36 | - |
| | | 35 | 16.60 | 0.30 | 13.37 | 0.96 | 1.27 | - |
| | 53.62°S, 36.34°W | 50 | 16.30 | 0.23 | 18.49 | 1.21 | 1.40 | - |
| | | 75 | 23.82 | 0.56 | 29.86 | 0.98 | 1.28 | - |
| | | 100 | 8.49 | 0.10 | 10.50 | 0.73 | 0.56 | - |
| | | 150 | 1.88 | 0.03 | 4.49 | 2.25 | 0.40 | - |
| | | 200 | 2.72 | 0.02 | 1.40 | 0.63 | 0.44 | 2.87 |





|  |  |  |  |  |  |  |  |
|---|---|---|---|---|---|---|---|
|  |  | 300 | 2.56 | 0.05 | 2.40 | 0.34 | 0.25 | - |
|  |  | 400 | 3.75 | 0.02 | 5.28 | 0.48 | 0.30 | 1.17 |
|  |  | 500 | 5.28 | 0.08 | 9.22 | 0.43 | 0.30 | - |
|  |  | 600 | 5.50 | 0.09 | 11.45 | 0.53 | 0.28 | 1.63 |
|  |  | 750 | 5.27 | 0.06 | 8.16 | 0.44 | 0.30 | - |
| 10/01/2011 | #17 (E133) | 20 | 10.92 | 0.22 | 7.43 | 2.31 | 1.20 | 3.76 |
|  |  | 35 | 20.83 | 0.53 | 16.22 | 1.81 | 1.34 | 2.56 |
|  | 53.90°S, 36.57°W | 50 | 34.59 | 1.00 | 57.55 | 2.29 | 1.42 | 2.33 |
|  |  | 75 | 118.25 | 2.18 | 64.36 | 4.21 | 1.86 | 2.19 |
|  |  | 100 | 50.71 | 1.00 | 77.52 | 2.48 | 1.42 | 1.62 |
|  |  | 150 | 112.28 | 2.23 | 86.09 | 3.39 | 1.41 | 0.86 |
| 11/01/2011 | #18 (E138) | 20 | 106.71 | 1.77 | 95.17 | 2.75 | 1.57 | 3.36 |
|  |  | 35 | 83.53 | 0.00 | 100.32 | 1.97 | 1.33 | 2.44 |
|  | 54.10°S, 36.25°W | 50 | 9.67 | 0.00 | 18.23 | 0.74 | 0.85 | - |
|  |  | 75 | 5.65 | 0.00 | 8.90 | 0.62 | 0.65 | - |
|  |  | 100 | 4.50 | 0.08 | 23.65 | 1.25 | 0.48 | 5.18 |
|  |  | 150 | 7.81 | 0.11 | 12.87 | 1.43 | 0.49 | 8.19 |
| 12/01/2011 | #19/20 (E141 & E143) | 20 | 60.19 | 2.11 | 54.29 | 1.46 | 1.71 | 5.30 |
|  |  | 35 | 60.17 | 2.19 | 87.17 | 1.34 | 1.90 | 8.22 |
|  | 53.54°S, 38.11°W | 50 | 66.78 | 2.74 | 141.75 | 1.57 | 1.90 | 8.73 |
|  |  | 75 | 71.69 | 1.78 | 79.19 | 1.61 | 2.13 | 11.45 |
|  |  | 100 | 10.77 | 0.25 | 32.12 | 0.99 | 0.67 | 10.74 |
|  |  | 150 | 5.43 | 0.13 | 31.35 | 1.84 | 0.92 | 12.00 |
|  |  | 200 | 7.92 | 0.14 | 27.42 | 1.45 | 0.60 | 9.60 |
|  |  | 400 | 5.35 | 0.00 | 23.61 | 1.61 | 0.45 | 18.44 |
|  |  | 600 | 5.81 | 0.10 | 35.99 | 1.06 | 0.38 | 10.74 |
|  |  | 800 | 4.26 | 0.13 | 35.67 | 1.07 | 0.36 | 11.95 |
| 13/01/2011 | #21 (E151) | 20 | 44.75 | 1.54 | 114.13 | 0.72 | 1.38 | 2.58 |
|  |  | 35 | 39.99 | 1.82 | 73.37 | 0.77 | 0.94 | 2.29 |
|  | 53.75°S, 38.98°W | 50 | 48.57 | 2.03 | 94.66 | 1.24 | 1.36 | 1.91 |
|  |  | 75 | 25.63 | 0.91 | 68.56 | 0.98 | 1.17 | - |
|  |  | 100 | 64.06 | 1.91 | 114.03 | 2.33 | 1.32 | 1.51 |
|  |  | 150 | 73.04 | 1.59 | 62.83 | 7.70 | 1.28 | 12.20 |






**Table 2:** Particulate Fe (PFe), Mn (PMn), and Al (PAl) concentrations in the top 150 m of
the water column at the 14 sites visited during JR247. The leachable particulate fraction (LP)
is indicated in percent. Additional information covers sampling date, station ID, event
number, latitude and longitude, and water column depth. (Depths marked by * indicate that
the polycarbonate filter was corrupted after retrieving the SAPS)

| Date | Station ID | Depth | Particulate (nmol L$^{-1}$) | | | Leach. Part. (%) | | |
|---|---|---|---|---|---|---|---|---|
| | Lat. & Lon. | (m) | PFe | PMn | PAl | LPFe | LPMn | LPAl |
| 25/12/2010 | #1/2 (E22) | 20 | 5.17 | 0.08 | 4.82 | 0.37 | 2.39 | 1.65 |
| | 53.70°S, 38.21°W | 50* | 9.12 | 0.14 | 7.91 | 0.27 | 2.61 | 1.47 |
| | (322 m) | 150* | 76.61 | 1.09 | 66.91 | 6.26 | 2.74 | 4.65 |
| 26/12/2010 | #3 (E31) | 20 | 6.62 | 0.09 | 6.64 | 0.02 | 3.30 | 0.79 |
| | 53.85°S, 39.14°W | 50 | 267.48 | 3.85 | 162.59 | 1.48 | 0.79 | 0.65 |
| | (287 m) | 150 | 4.36 | 0.06 | 4.26 | 0.07 | 1.55 | 1.93 |
| 31/12/2010 | #4/5 (E72) | 20 | 8.52 | 0.12 | 7.99 | 0.51 | 1.68 | 2.62 |
| | 53.49°S, 37.71°W | 50 | 15.15 | 0.23 | 12.96 | 0.56 | 2.44 | 2.74 |
| | (1917 m) | 150 | 2.33 | 0.03 | 2.15 | 0.65 | 1.78 | 2.42 |
| 02/01/2011 | #6 (E80) | 20 | 85.74 | 1.11 | 59.05 | 1.60 | 2.28 | 4.50 |
| | 53.99°S, 36.37°W | 50 | 17.76 | 0.24 | 8.87 | - | - | - |
| | (208 m) | 150 | 137.39 | 2.02 | 98.54 | 3.46 | 0.91 | 2.81 |
| 03/01/2011 | #7/8 (E88) | 20 | 1.95 | 0.02 | 0.87 | 0.13 | 2.97 | 4.99 |
| | 54.10°S, 35.46°W | 50 | 1.67 | 0.02 | 0.92 | 0.08 | 4.35 | 4.24 |
| | (330 m) | 150 | 1.23 | 0.02 | 0.71 | 0.19 | 2.11 | 5.13 |
| 04/01/2011 | #9/10 (E96) | 20 | 20.91 | 0.08 | 15.74 | 0.56 | 5.01 | 3.24 |
| | 54.26°S, 35.35°W | 50 | 19.16 | 0.27 | 15.58 | 0.45 | 1.22 | 2.51 |
| | (263 m) | 150 | 54.06 | 0.77 | 48.10 | 1.08 | 1.65 | 2.08 |
| 05/01/2011 | #11/12 (E100) | 20* | 1.49 | 0.01 | 0.86 | 0.18 | 4.42 | 2.92 |
| | 54.62°S, 34.81°W | 50 | 0.87 | 0.01 | 0.60 | 0.27 | 6.63 | 4.20 |
| | (747 m) | 150 | 1.76 | 0.03 | 1.08 | 0.37 | 4.38 | 3.33 |
| 06/01/2011 | #13 (E106) | 20 | 2.75 | 0.03 | 1.78 | 0.63 | 3.13 | 4.29 |
| | 54.53°S, 35.27°W | 50 | 4.11 | 0.05 | 3.07 | 0.44 | 2.04 | 2.76 |
| | (133 m) | 100 | 10.28 | 0.15 | 7.62 | 0.46 | 1.70 | 2.54 |
| 07/01/2011 | #14 (E114) | 20 | 2.80 | 0.04 | 1.84 | 0.07 | 1.58 | 3.29 |
| | 54.56°S, 35.59°W | 50 | 1.41 | 0.02 | 0.97 | 0.10 | 2.57 | 3.92 |
| | (263 m) | 150 | 31.34 | 0.46 | 26.92 | 0.72 | 1.57 | 2.28 |
| 08/01/2011 | #15/16 (E120) | 20 | 24.54 | 0.37 | 22.91 | 0.85 | 3.95 | 1.88 |
| | 53.62°S, 36.34°W | 50 | 27.72 | 0.40 | 23.23 | 0.43 | 3.65 | 1.36 |
| | (852 m) | 150 | 4.74 | 0.07 | 3.94 | 0.90 | 4.31 | 1.06 |
| 10/01/2011 | #17 (E134) | 20 | 10.43 | 0.14 | 8.09 | 0.34 | 1.66 | 2.41 |
| | 53.90°S, 36.57°W | 50 | 43.04 | 0.60 | 38.79 | 1.34 | 1.07 | 1.67 |
| | (209 m) | 150 | 207.48 | 3.10 | 194.88 | 1.72 | 0.82 | 1.50 |
| 11/01/2011 | #18 (E139) | 20 | 95.52 | 1.32 | 88.39 | 1.39 | 1.82 | 1.93 |





| | | | | | | | |
|---|---|---|---|---|---|---|---|
| | 54.10°S, 36.25°W | 50 | 37.43 | 0.52 | 35.33 | 1.16 | 1.29 | 1.85 |
| | (276 m) | 150 | 28.00 | 0.41 | 23.60 | 1.26 | 2.35 | 2.27 |
| 12/01/2011 | #19/20 (E142) | 20 | 97.60 | 1.52 | 97.10 | 0.16 | 1.66 | 0.33 |
| | 53.54°S, 38.11°W | 50 | 90.96 | 1.42 | 92.89 | 0.39 | 1.98 | 0.80 |
| | (1741 m) | 150 | 7.41 | 0.12 | 6.37 | 0.74 | 8.25 | 2.75 |
| 13/01/2011 | #21 (E152) | 20 | 50.75 | 0.85 | 52.78 | 0.06 | 2.99 | 0.12 |
| | 53.75°S, 38.98°W | 50 | 59.59 | 0.93 | 59.98 | 0.05 | 2.15 | 0.09 |
| | (269 m) | 150 | 153.48 | 2.34 | 89.63 | 3.14 | 1.10 | 2.94 |





**Table 3:** Particulate iron (SFe), aluminum (SAl), and manganese (SMn) concentrations in
shelf sediments collected during JC055 in January and February 2011. Pore water data
retrieved additionally from these three cores are listed for Fe (Fe$_{PW}$) and Mn (Mn$_{PW}$).
Additional information are event number (MC…), latitude and longitude, and water column
depth.

| Station ID<br>Lat. & Lon. | Depth<br>(cm) | SFe<br>(mol kg-1) | SAl<br>(mol kg-1) | SMn<br>(mmol kg-1) | Fe$_{PW}$<br>(μmol kg-1) | Mn$_{PW}$<br>(μmol kg-1) |
|---|---|---|---|---|---|---|
| #S1 (MC33) | 0.5 | 0.58 | 1.77 | 11.56 | 3.01 | 2.29 |
| 54.16°S, 37.98°W | 1.5 | 0.61 | 1.74 | 11.52 | 17.47 | 0.84 |
| (257 m) | 2.5 | 0.59 | 1.77 | 11.78 | 110.90 | 0.28 |
| | 3.5 | 0.6 | 1.86 | 12.05 | 106.24 | 0.53 |
| | 4.5 | 0.58 | 1.72 | 11.82 | 94.09 | 0.34 |
| | 5.5 | 0.59 | 1.86 | 12.04 | 82.79 | 0.27 |
| | 9 | 0.56 | 1.72 | 11.19 | 32.98 | 0.00 |
| | 15 | 0.55 | 1.74 | 11.15 | 2.44 | 0.06 |
| | 25 | 0.53 | 1.6 | 10.81 | 0.80 | 0.16 |
| #S2 (MC34) | 0.5 | 0.64 | 1.77 | 11.42 | 1.53 | 0.87 |
| 54.16°S, 37.94°W | 1.5 | 0.6 | 1.79 | 11.73 | / | / |
| (247 m) | 2.5 | 0.58 | 1.76 | 11.81 | 0.97 | 0.24 |
| | 6.5 | 0.59 | 1.83 | 12.23 | 11.19 | 0.26 |
| | 10.5 | 0.58 | 1.8 | 11.78 | 14.28 | 0.25 |
| | 14.5 | 0.54 | 1.6 | 10.83 | 3.59 | 0.33 |
| | 16.5 | 0.56 | 1.72 | 11.22 | 2.27 | 0.31 |
| #S3 (MC35) | 0.5 | 0.61 | 1.67 | 11.42 | 1.46 | 0.43 |
| 54.15°S, 37.97°W | 1.5 | 0.59 | 1.76 | 11.7 | 28.94 | 0.35 |
| (254 m) | 2.5 | 0.58 | 1.76 | 11.7 | 91.52 | 0.37 |
| | 3.5 | 0.59 | 1.81 | 12.03 | 40.16 | 0.44 |
| | 5.5 | 0.57 | 1.78 | 11.58 | 49.37 | 0.56 |
| | 8.5 | 0.59 | 1.82 | 11.65 | 67.92 | 0.52 |
| | 17 | 0.54 | 1.69 | 10.8 | 3.87 | 0.34 |
| | 19 | 0.55 | 1.67 | 10.86 | 1.82 | 0.12 |
| | 25 | 0.55 | 1.77 | 11.19 | 2.73 | 0.36 |
| | 29 | 0.56 | 1.79 | 11.19 | 5.64 | 0.16 |





**Table 4:** Total and leachable particulate Fe, Mn, and Al determined for the 27 individual krill faecal pellet samples collected during 9 krill incubation experiments on-board RRS *James Clark Ross* (JR247).

| # Sample | pellet weight (mg) | Total Fe ($\mu$g mg$^{-1}$) | Total Al ($\mu$g mg$^{-1}$) | Total Mn (ng mg$^{-1}$) | Leach. P. Fe (%) | Leach. P. Al (%) | Leach. P. Mn (%) |
|---|---|---|---|---|---|---|---|
| 1 | 4.87 | 0.88 | 1.06 | 12.5 | 6.33 | 8.83 | 13.24 |
| 2 | 2.18 | 1.33 | 1.68 | 16.7 | 3.02 | 8.81 | 8.22 |
| 3 | 4.26 | 1.07 | 1.90 | 17.8 | 5.37 | 3.27 | 11.81 |
| 4 | 1.91 | 5.19 | 5.53 | 76.1 | 2.15 | 1.95 | 5.68 |
| 5 | 1.41 | 2.70 | 2.84 | 39.1 | 2.46 | 1.59 | 3.54 |
| 7 | 7.80 | 67.1 | 64.2 | 998.3 | 2.93 | 2.21 | 3.25 |
| 8 | 0.99 | 2.71 | 2.42 | 35.0 | 3.76 | 4.59 | 5.99 |
| 10 | 1.48 | 6.42 | 4.89 | 71.6 | 0.29 | 4.83 | 0.91 |
| 13 | 2.79 | 4.13 | 3.11 | 50.3 | 0.36 | 5.07 | 1.53 |
| 15 | 0.77 | 37.3 | 38.1 | 531.1 | 2.03 | 2.80 | 6.21 |
| 16 | 1.21 | 6.35 | 6.22 | 81.2 | 1.24 | 7.47 | 3.13 |
| 18 | 12.27 | 40.0 | 36.6 | 582.5 | 3.95 | 2.07 | 4.29 |
| 19 | 2.19 | 11.2 | 9.49 | 146.9 | 0.15 | 2.03 | 1.07 |
| 22 | 2.43 | 48.1 | 49.7 | 721.5 | 0.81 | 2.32 | 0.98 |
| 40 | 3.35 | 22.8 | 22.0 | 337.4 | 5.51 | 3.21 | 5.50 |
| 41 | 8.55 | 6.91 | 7.14 | 103.1 | 1.11 | 1.88 | 4.31 |
| 42 | 3.5 | 25.7 | 24.8 | 376.2 | 5.09 | 2.98 | 5.29 |
| 45 | 0.40 | 3.96 | 4.43 | 43.3 | 1.27 | 13.90 | 1.46 |
| 47 | 7.65 | 3.63 | 3.92 | 52.7 | 0.34 | 0.68 | 3.65 |
| 48 | 0.63 | 3.06 | 3.21 | 34.1 | 0.05 | 4.22 | 0.76 |
| 49 | 4.42 | 29.6 | 28.5 | 438.4 | 1.65 | 2.93 | 1.95 |
| 50 | 7.46 | 2.31 | 2.37 | 34.6 | 0.36 | 0.51 | 2.78 |
| 51 | 5.18 | 28.0 | 27.1 | 431.3 | 1.85 | 2.60 | 2.01 |
| 62 | 1.20 | 4.63 | 4.68 | 68.0 | 0.31 | 1.78 | 0.47 |
| 68 | 2.25 | 44.0 | 40.2 | 667.4 | 4.84 | 1.95 | 4.77 |
| 69 | 1.66 | 43.6 | 44.8 | 663.7 | 5.66 | 2.13 | 5.46 |
| 71 | 3.47 | 35.3 | 36.4 | 557.7 | 1.50 | 1.99 | 1.76 |




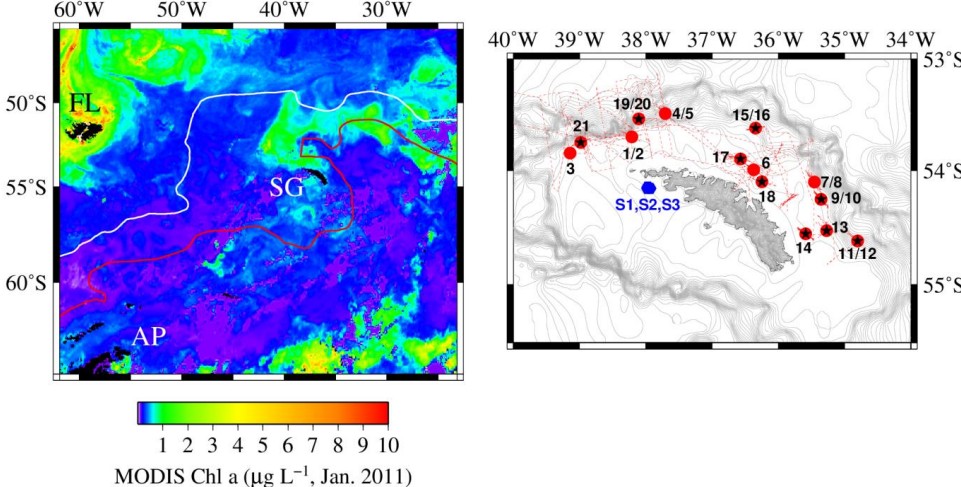


**Figure 1:** (Left figure) Locations of Falkland Islands (FL), South Georgia (SG), and

Antarctic Peninsula (AP) in the Atlantic sector of the Southern Ocean. South Georgia is

located between the Antarctic Polar Front (PF, white line) and the Subantarctic Circumpolar

Current Front (SACCF, red line). The colour bar represents the Chlorophyll a (Chl a) content

recorded by the MODIS satellite in January 2011. (Right figure) The region around SG and

the OTE (black stars) and SAPS sampling sites (red points) visited during JR247. The red

dashed line illustrates the cruise track of JR247. The three sediment sampling sites S1, S2,

and S3 visited during JC055 are shown by blue hexagons. The ocean bathymetry of the

region was plotted using the GEBCO bathymetric data set. The shelf of South Georgia is

between 100 and 250 m deep and extends about 30 to 100 km (shelf edge indicated by high

density of isobaths).







**Figure 2:** (Upper row) Distribution of particulates in unfiltered seawater samples ($LP_{Un}Fe$ in
black), manganese ($LP_{Un}Mn$ in blue), and aluminium ($LP_{Un}Al$ in red) in the water column of
stations located on the island shelf (125 m – 270 m water depth). The particulate Fe (PFe)
fraction retrieved by SAPS is illustrated with open black circles and corresponds to the
concentration labels of $LP_{Un}Fe$. Concentrations above 120 nmol $L^{-1}$ are listed in Table 1 and
2. Error bars represent the standard deviation of the analysis. Density sigma-theta ($\sigma_0$) in kg
$m^{-3}$ is illustrated by the black dashed line. (Lower row) Dissolved iron (DFe), manganese
(DMn), and aluminium (DAl) are represented by the same colour code as above. Dashed
lines illustrate Chlorophyll a (Chl a) content of the water column recorded by the CTD
fluorometer.





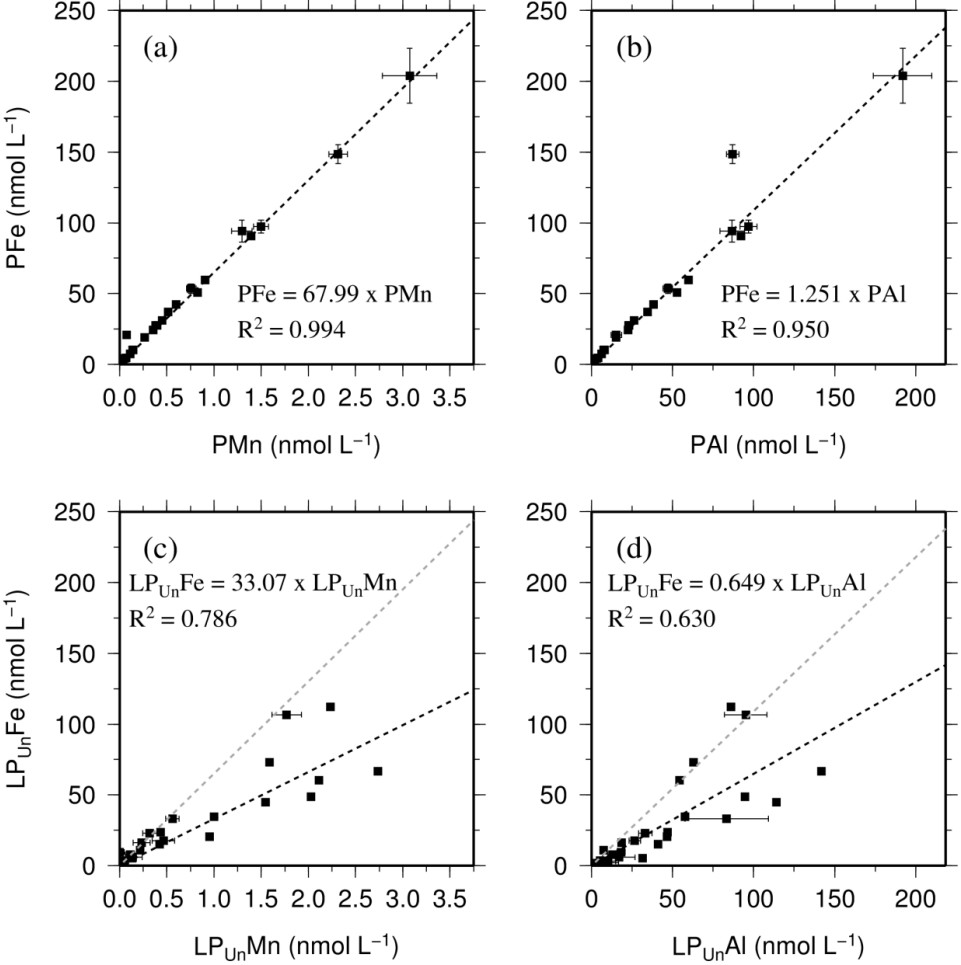


**Figure 3:** Relationship of the entire data set for the particulate fraction of Fe, Mn, and Al in

particulates (P) retrieved using SAPS ((a) and (b)) and the leachable particulate fraction

($LP_{UN}$) estimated from unfiltered and dissolved seawater samples collected using OTE bottles

((c) and (d)). Error bars represent the standard deviation of the analysis. The linear regression

of each relationship is illustrated by a dashed black line, the formula, and the $R^2$.  The grey

dashed line in c. and d. represents the linear relationship of particulate trace meals (P) shown

in (a) and (b).

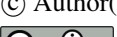


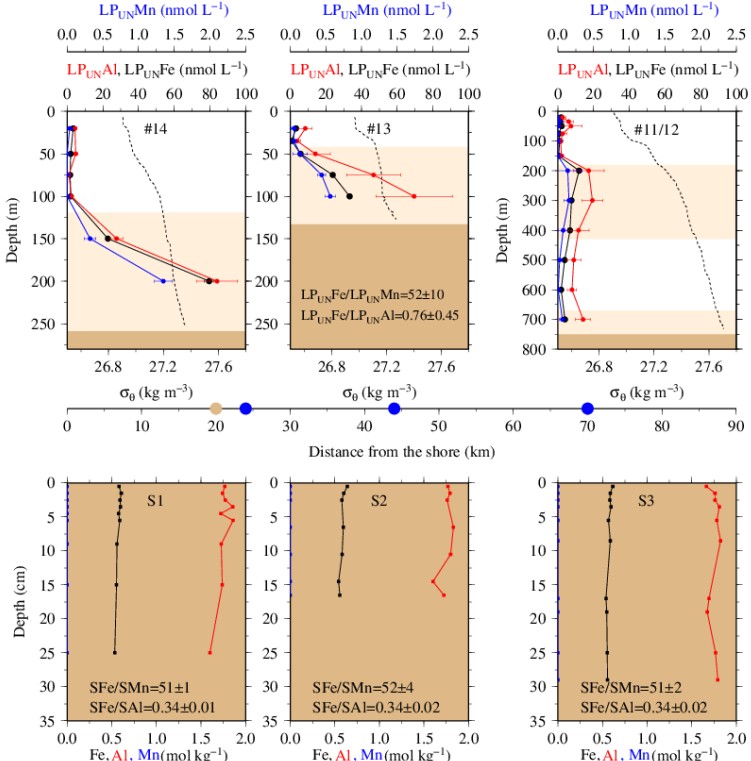

**Figure 4:** (Upper row) From left to right, concentrations of leachable particulate iron

(LP$_{Un}$Fe), aluminium (LP$_{Un}$Al), and manganese (LP$_{Un}$Mn) of unfiltered seawater samples for

the two shelf stations #14, #13 and the shelf edge station #11/12 (Note different depth

scaling). Error bars represent the standard deviation of the analysis. Water density (sigma-

theta ($\sigma_0$)) is shown by the dashed black line. Brown areas represent sediments and pink areas

the zone of resuspended sediment particles in the water column. Diagram 14 (left) contains

the average LP$_{Un}$Fe/LP$_{Un}$Al and LP$_{Un}$Fe/LP$_{Un}$Mn ratio of particles in seawater samples

collected within the pink layers. (Lower row) Diagram S1, S2 and, S3 displays the Fe, Mn,

and Al content in the three sediment cores. Shown are average SFe/SAl and SFe/SMn ratios

(mol/mol) of particles from the surface layer for station S1, S2, and S3. Dots on the distance

scaling in the middle represent the distance of each water column station (blue) and sediment

core (brown) station to the nearest shore.

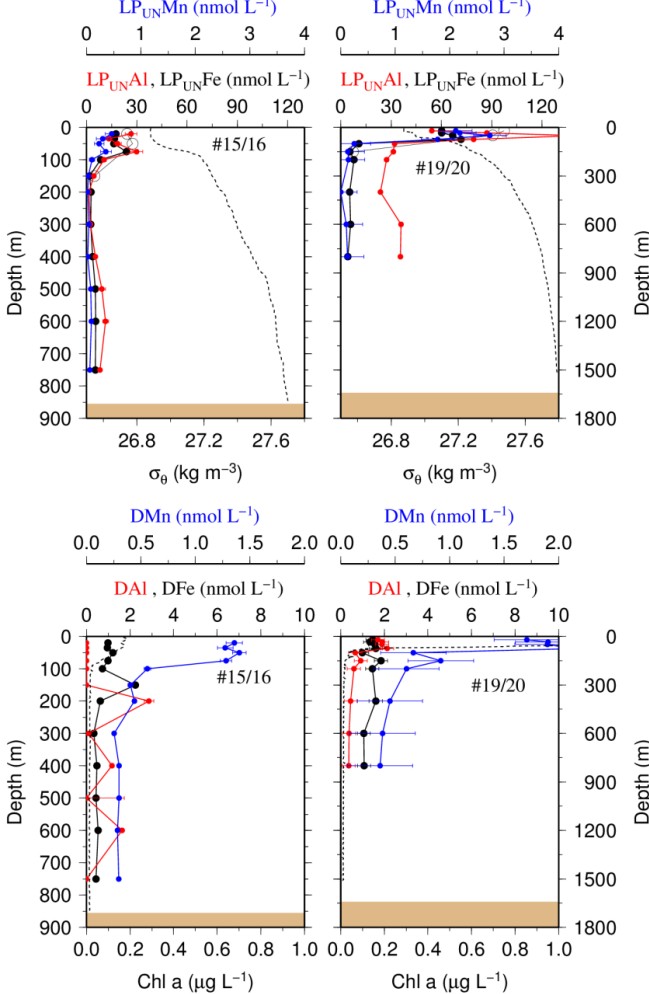

858

**Figure 5:** (Upper row) Distribution of leachable particulate manganese (LP$_{Un}$Mn in blue),

iron (LP$_{Un}$Fe in black), and aluminium (LP$_{Un}$Al in red) concentrations in the water column of

the two other stations located on the island shelf edge (> 700 m water depth). The particulate

Fe (PFe) is illustrated by black circles and corresponds to the concentration labels of LP$_{Un}$Fe.

Error bars represent the standard deviation of the analysis. Sigma-theta ($\sigma_\theta$) is illustrated by

the black dashed line. (Lower row) Dissolved manganese (DMn), iron (DFe), and aluminium

(DAl) are represented by the same colour code as for the upper row. Dashed line illustrates

the Chl a content of the water column recorded by the CTD mounted fluorometer.



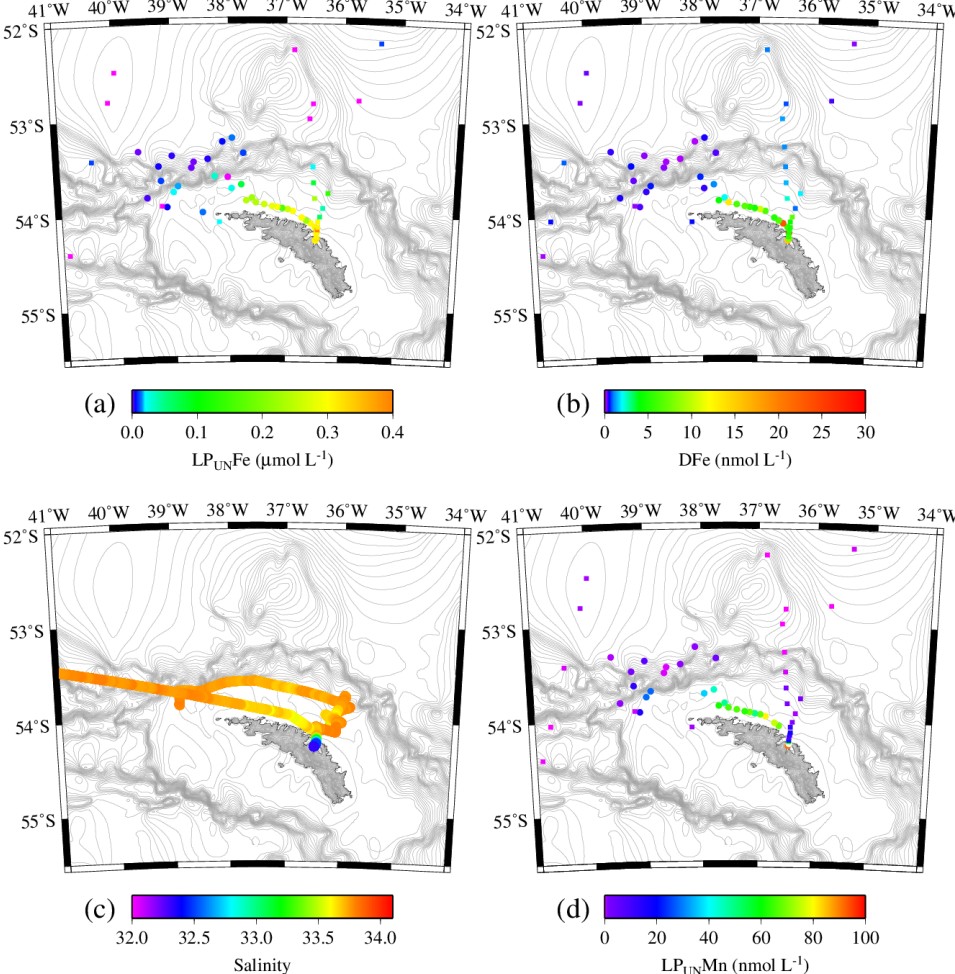

867

**Figure 6:** Concentrations of leachable particulate Fe (LP$_{UN}$Fe) of unfiltered seawater samples

(a), dissolved Fe (DFe) (b), Salinity (c) and leachable particulate Mn (LP$_{UN}$Mn) in unfiltered

seawater samples (d) in surface waters collected during JR247 (circles) and JR274 (squares)

around South Georgia. Isobath are represented by grey lines (GEBCO – Gridded Bathymetry

Data).



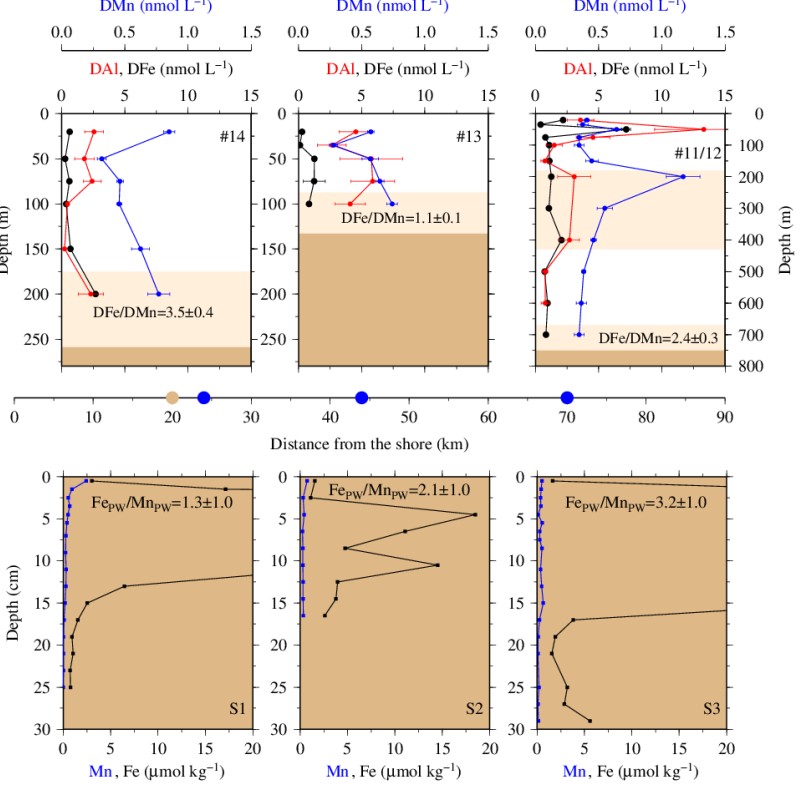

**Figure 7:** (Upper row) From left to right, concentrations of dissolved iron (DFe), aluminium

(DAl), and manganese (DMn) for the two shelf stations (#14, #13) and the shelf edge station

(#11/12). Note different depth scaling. Error bars represent the standard deviation of the

analysis. Pink areas represent the zone of resuspended sediments in the water column. The

DFe/DMn ratios of the seawaters collected within the pink zone is indicated. (Lower row)

Diagram S1, S2 and, S3 displays the Fe (black), and Mn (blue) content in pore waters of the

three sediment cores. Values off-axis can be found in Table 3. Shown are average

$Fe_{PW}/Mn_{PW}$ ratios (mol/mol) of top surface layer (1 cm) for station S1, S2, and S3. Dots on

the distance scaling in the middle represent the distance of each water column station (blue)

and sediment core (brown) station to the nearest shore.



884

**Figure 8:** Sketch of DFe fluxes on the shelf, in the transition zone and in the downstream

blooming region seprated by the red dashed lines. (left sketch) describes the dissolved Fe

fluxes on the shelf that together generate Fe rich biogenic and lithogenic particles (dark

green). These are transferred offshore (light green arrows) following the ACC to open ocean

sites (sketch in the middle). Iron enriched particles (dark green suns) in the transition zone

are recycled and supplement DFe requirements of the phytoplankton community in the

transition zone. During each cycle of recycling and uptake an unknown Fe fraction is lost by

vertical export. (right sketch) describes the dissolved Fe fluxes in the blooming zone.