# Peer review of "Mechanisms of dissolved and labile particulate iron supply to shelf waters and phytoplankton blooms off South Georgia, Southern Ocean"

_Biogeosciences, 2017_

## Referee Comment (RC1) · Anonymous Referee #1 · 6 Nov 2017

General Comments: This manuscript presents new dissolved, particulate, sediment, and porewater Fe, Mn, and Al data from the South Georgia Island region. The authors conclude that "The majority of the Fe appears to be derived from remineralisation of Fe enriched phytoplankton cells/biogenic particles that are transported with the water masses into the bloom region". However, this conclusion is based on a Fe budget that is extremely poorly constrained, so it should be presented as a hypothesis, rather than a conclusion supported by their measurements. Overall, this is a useful dataset to add to the quickly growing datasets of Fe in naturally fertilized regions in the Southern Ocean. There are, however, many aspects in this particulate dataset that aren't consistent, at least in the authors' discussion of it. Sections 3.1.1 and 3.1.2 need to be overhauled

(detailed comments below). There are several other inconsistencies: for example, assuming the region is upwelling for the calculation of benthic Fe flux, but downwelling for calculation of the Fe budget. An attempt is made at a horizontal transport term for the PFe, but then dropped from the budget. This paper needs major revisions before it can be published.

Specific comments: Methods:

-Why were sediment cores collected on the southern shelf and not the northern shelf? Given that the phytoplankton bloom appears to original on the northern shelf, and all water column and particle samples were collected on the northern shelf, this seems like a strange decision. If this was outside of the authors' control, then this needs to at least be discussed when sediments are compared to the water column, since one would expect the stronger bloom on the northside to affect the benthic processing of Fe and Mn, so sediments on the southern shelf may not be representative of those on the northern shelf.

-Fecal pellet data are first discussed in section 3.1.3 (for data in Table 4) but without introducing the methodology for fecal pellet sampling and analysis. They are discussed again in section 3.2.3, where a reference is to the Schmidt et al papers, but this should be moved up.

Section 3.1.1

-The discussion of particulate fractions in this section is confusing. LPun is derived from acidified unfiltered seawater and defined in equation 1, but then is compared to P as if they should be the same, differing only in their sampling methods (acidified un-filtered seawater vs SAPS sample—lines 157-158). However, P is defined in section 2.2 as the sum of leachable (LP) and refractory (RP) fractions from the SAPS samples, so one wouldn't expect LPun and P to be the same. I initially read this section thinking that they were comparing LPun (from acidified unfiltered seawater) and LP (from acetic acid leach of SAPS samples), which is the more direct comparison if they want

to isolate sampling differences (but still not perfect since acidification to pH 1.7 with HNO3 is still not the same as a 25% acetic acid leach, but at least more comparable). This section needs to be clarified. It seems that they have two points in this section: 1) that pFe and pAl have a refractory component, since LP(un)/P < 100%, and 2) LP(un) scales with P, so they want to justify using LP(un) for P. For the first point, it seems that a wholly SAPS-derived assessment of LP/(LP+RP) would be the better parameter to present, because then there is no confusion of mixing sampling systems, pore sizes, and leach types. This is done in section 3.2.2 (lines 320-340), which would make more sense in this section

-For the second point, presumably, they want to use the bottle-derived leachable particulate data because of higher depth resolution than SAPS samples, so this should be a separate argument than the one about the presence of refractory Fe and Al.

-Table 2: are the LPFe, LPAl, LPMn columns derived from the SAPS LP or the bottle LPun? This should be specified in the caption. The units for the LP are specified as being in percent. But these values are all very small—mostly less than 5%, even for Mn, which contradict what was stated in the text in lines 161-163: "The LPUn corresponded to ca. 63±4% of the PFe, 83±11% of the PAl and 100±10% of the PMn fractions." Presumably, this discrepancy arises because the Table 2 values are SAPS-based (LP/P), whereas the values in the text are mixing and matching bottle-based LPun and SAPS-based totals. Is that correct? If this is the case, then this means that the bottle-based LPun are much higher than the SAPS-based LPs. One might expect this to some degree, since the LPun is acidified to a slightly lower pH (∼1.7) and for a lot longer than the SAPS-based LPs (25% acetic acid should have a pH ∼ 2.1), but I wouldn't have expected the difference to be so big. This suggests that the particle population collected by the SAPS may be a subset of the particles accessed by the LPun method, and calls into question whether normalizing or comparing the bottle based LPun by/to the SAPS-based P is appropriate or meaningful. As further detailed below, I suggest that the authors do not mix bottle-based and SAPS-based parameters (i.e.

they should not report or interpret LPun/P).

Section 3.1.2: this section needs to be considerably revised to account for the following:

-Re. lithogenic source of suspended particles: The extremely good correlations of the PFe to PMn and PFe to PAl concentrations (Fig 3a,3b) do support their contention that the particles have a single lithogenic origin (line 197), since biogeochemical processing of the particles would be likely to affect Mn, Fe, and Al differently, and therefore cause more scatter in the data. However, the slopes of their relationships don't support this. The authors base the conclusion that the suspended particles have a lithogenic origin on the slope of the PFe to PMn data (68 mol Fe/mol Mn), which they say agrees well with a typical crustal ratio (they use 60 mol Fe/mol Mn from Wedepohl, which is close to 57 and 50 mol Fe/mol Mn for UCC and BCC crustal averages, respectively, reported by (Taylor and McLennan, 1995)). However, PMn is not usually a good crustal indicator, since Mn-oxides are frequently a large component of marine particulate Mn. For this reason, PAl or PTi are more frequently used to assess how lithogenic the PFe is. However, their PFe to PAl relationship (slope=1.251 mol Fe to mol Al) far exceeds the slope of typical crust (UCC Fe:Al=0.21; BCC Fe:Al=0.41 (Taylor and McLennan, 1995)). Since the sediment elemental ratios from the southern shelf (Table 3, line 199) are close to these crustal averages, this suggests a fairly large component of the PFe that is in excess of a lithogenic source. One can derive a PMn to PAl ratio from their data=PFe:PAl / PFe:PMn = 0.018 mol Mn/mol Al, which is also greater than typical crustal ratios (UCC Mn:Al=0.0037; BCC Mn:Al=0.0082). So just comparing their slopes to crustal ratios to Al, one would expect there to be a fair amount of non-lithogenic Fe and Mn assuming crustal averages are a reasonable approximation of the local sources (a plausible assumption given that the sediment elemental ratios are close to crustal averages). The conclusion that the suspended particles are primarily lithogenic is therefore not supported by their data.

-The slopes of Fe:Mn and Fe:Al of the LPun fractions are each about half of those for the total particulate fractions (Fig 3a-d). Comparing the leachable and total slopes,

this suggests that particulate Fe is about half as labile as particulate Mn or particulate Al. This is much more consistent with their wholly SAPS-based %leachable data from Table 2 (average %leachable Fe < average %leachable Mn∼average %leachable Al), than with their LPun/P from lines 161-163 (Fe<Al<Mn), another reason not to mix the bottle-based (LPun) and SAPS-based (LP, P) parameters.

-To rule out data quality issues, the authors should report results from an external standard such as a standard reference material that was run alongside the digestion of the suspended particle leaches/digests. External standards in the supplements were only reported for seawater and for sediments, which used a different digestion procedure than the particles. Was there an external standard measured for the suspended particles to ensure that the digestion method was working accurately?

Section 3.1.3

-Re. meltwater source of LPunFe: is there a relationship between LPunFe and salinity? A scatter plot of these parameters would be evidence of a meltwater source of LPunFe.

-Lines 253-256: PFe/PAl in krill fecal pellets (0.48) is much lower than PFe/PAl in suspended particles (1.25—Fig 3b), though it is relatively close to the sediment ratios from the southern shelf. This suggests that the suspended particles have an additional source of PFe compared to krill or sediments.

Section 3.2.1

-The authors should acknowedge that the sediment core data were taken on the non-productive, southern side of the island, whereas the water column data were taken from the productive, northern side of the island, so benthic fluxes calculated from porewater profiles could be rather disconnected to the measured DFe.

-The authors calculate a vertical supply of DFe to surface mixed layers by assuming a vertical diffusivity and vertical advection (upwelling) used by De Jong et al. 2012 for the region downstream of the Antarctic Peninsula. But later in section 3.4.2, they discuss

a study by Tagliabue et al. 2014 modelling DFe supply for South Georgia, in which Ekman \*downwelling\* (not upwelling!) prevailed, removing DFe from the surface, rather than supplying it. I can appreciate that there's uncertainty in the estimate of vertical advection, but they should probably pick a sign and stick to it!

Section 3.3: there are some problems with the proposed flowpaths:

-Line 384: I do not understand the proposed advective pathway: the authors reference "Fig. 1; #11/12 via #13 to #14" as a NE-SW transect. First, these stations do not define a NE-SW transect. Second, what is the order of the proposed advective pathway? Is it 13 to 14 to 11/12, even though 13 is actually further from the island than 14? Is 13 shallower than 14 (Fig 4) despite being further offshore (Fig 1)?

-Presuming that the authors are assuming that #11/12 is the offshore end of the flow-path, this is inconsistent with DMn, which is higher at 11/12 compared to 13 and 14. It's no wonder that they were not able to fit an exponential function to the DFe data (lines 409-410). If this isn't an advective flowpath, the good exponential fit to the LPUnFe data may be a coincidence, or at least unrelated to offshore transport. Further, the "exponential decrease" in PFe was based on a 2 point fit! No wonder $R^2=1$! This should be removed.

Section 3.4: budget

-As noted above, there is a discrepancy in the manuscript about whether this region is downwelling, as assumed here for the budget, or upwelling, as assumed in section 3.2.1—sediment DFe flux.

-In the manuscript, an overall vertical loss of DFe of -0.0025 umol DFe/m2/d is assumed (line 465), but in Figure 8, a vertical supply of +0.009 umol DFe/m2/d is indicated.

-Given the poor (lack of) constraints on both horizontal and vertical supply of DFe, the suggestion that there is a mismatch in the supply and demand that is filled by advecting biogenic Fe is a nice hypothesis, but rather speculative, both the size of the term, and

the nature of it. The size of the flux is not constrained, since they did not have local estimates of horizontal or vertical supply. Re. the nature of their proposed flux: their PFe:PAl ratio from earlier in the manuscript does suggest excess particulate Fe over lithogenic, so this is at least consistent with an available biogenic Fe pool on the shelf, but is also consistent with Fe oxyhydroxides (i.e., authigenic, not biogenic particulate Fe).

Miscellaneous:

-Lines 104-107: I assume that unfiltered samples were also collected from the tow fish in JR247, since LPun values are plotted for this cruise in figure 6. Please add this to this section.

-Line 231-2: Fig 6a shows that surface concentrations of 0.3 $\mu$M LPunFe are quite common on the shelf, but 22 uM doesn't show up in this figure, so referring to it here is not that helpful.

Typos: Line 106: "cartridge filter", not "filer" Line 178: "where", not "were" Line 251: "ascent" not "ascend" Line 415: add "there was" to "...that *there was* a limited transfer..." Line 459: "Tagliabue" not "Tagliabu"

References

Taylor, S.R., McLennan, S.M., 1995. The Geochemical Evolution of the Continental-Crust. Reviews of Geophysics 33 (2), 241-265.

---

## Author Comment (AC1) · 21 Nov 2017

We thank the reviewer for their comments, but think that important findings of this paper were misunderstood or overlooked. For that reason we just conducted minor changes. This paper continuous the recently published paper from Schmidt et al. [2016] and highlights the strength of the Fe supply from the South Georgia shelf by applying three independent data sets collected in 2011 and 2013 on the shelf and shelf edge of South Georgia. The supply of dissolved Fe form shelf sediments, glacial outflow and zoo-plankton activity on the shallow shelf is discussed in section 3.1, 3.2 and 3.3. Section 3.4 discusses the supply of DFe in the bloom region north of the island, ca. 1,250 km

downstream the South Georgia source region. This is stated in the paragraph, line 429ff. Oceanographic settings at our study site on the South Georgia shelf (source region) are entirely different to open ocean sites downstream the island (Fe budget in the bloom region). Vertical diffusive and advective mixing are much stronger on the shelf than off-shore. For instance, internal waves that cross the 200 m deep shelf create strong turbulences that facilitate the diffusive and advective term of vertical mixing (upwelling, references are listed below). Father off shore in the bloom region, the vertical Fe supply is limited because of a deep ferrocline, reduced diffusive mixing and negative Eckman transport (downwelling [Tagliabue et al., 2014]). Recent publications illustrate that DFe supplied from continental or island arc shelf sediments drive offshore primary productivity [Graham et al., 2015]. The intriguing question remains, how DFe from the source region (continental shelf and island arc) is transported downstream in the region with elevated photosynthetic activity (bloom region). This shelf or island effect can be found on numerous islands in the Southern Ocean [Graham et al., 2015]. Although we mainly rely on literature values for the bloom region we estimated that the Fe demand in the bloom region is higher than the actual Fe supply (section 3.4). We suggested that the missing DFe quote must be delivered from biogenic Fe particles that were created under high Fe conditions on the South Georgia shelf and Fe was released downstream the island by internal recycling in the mixing layer. This process has been discussed in detail line 516ff. Please find the comments of the reviewer, and our reply to it underneath (Answer:....).

Specific comments: Methods: -Why were sediment cores collected on the southern shelf and not the northern shelf? Given that the phytoplankton bloom appears to original on the northern shelf, and all water column and particle samples were collected on the northern shelf, this seems like a strange decision. If this was outside of the authors' control, then this needs to at least be discussed when sediments are compared to the water column, since one would expect the stronger bloom on the northside to affect the benthic processing of Fe and Mn, so sediments on the southern shelf may not be representative of those on the northern shelf. Answer: We agree that sediment

cores from the productive northern side of the island would be better. However, JC055 was a benthic cruise, hunting gas hydrates, and thus sampling locations were outside our control. So, you can't get always what you want! We agree that an elevated organic content in marine sediments would alter substantially benthic redox processes and usually increases the content of Fe(II) and Mn(II) in pore waters and thus facilitates the efflux of sedimentary Fe. This may mean that the bentic Fe supply on the northern side is probably higher and the efflux on the southern side (0.1 – 44 $\mu$mol m-2 d-1) underestimates the northern shelf efflux. However, this has no effect on our vertical Fe supply calculations in surface waters on the northern side (see supplementary text). To clarify this issue we added a sentence in the section 2.1 (line 117ff) and section 3.2.1 (line 291ff).

-Fecal pellet data are first discussed in section 3.1.3 (for data in Table 4) but without introducing the methodology for fecal pellet sampling and analysis. They are discussed again in section 3.2.3, where a reference is to the Schmidt et al papers, but this should be moved up. Answer: To clarify we added "as described by Schmidt et al. [2016]" (line 254ff) and added "and fecal pellets" to the header of section 2.2 (line 126).

Section 3.1.1 -The discussion of particulate fractions in this section is confusing. LPun is derived from acidified unfiltered seawater and defined in equation 1, but then is compared to P as if they should be the same, differing only in their sampling methods (acidified unfiltered seawater vs SAPS sampleâËŸAËǦ Tlines 157-158). However, P is defined in section 2.2 as the sum of leachable (LP) and refractory (RP) fractions from the SAPS samples, so one wouldn't expect LPun and P to be the same. I initially read this section thinking that they were comparing LPun (from acidified unfiltered seawater) and LP (from acetic acid leach of SAPS samples), which is the more direct comparison if they want to isolate sampling differences (but still not perfect since acidification to pH 1.7 with HNO3 is still not the same as a 25% acetic acid leach, but at least more comparable). This section needs to be clarified. It seems that they have two points in this section: 1) that pFe and pAl have a refractory component, since LP(un)/P < 100%,

and 2) LP(un) scales with P, so they want to justify using LP(un) for P. For the first point, it seems that a wholly SAPS-derived assessment of LP/(LP+RP) would be the better parameter to present, because then there is no confusion of mixing sampling systems, pore sizes, and leach types. This is done in section 3.2.2 (lines 320-340), which would make more sense in this section Answer: We disagree with the reviewer. We observed that Iron, Mn, and Al in the LPUn and P fraction show a linear relationship (see Fig. 3 and Table 1and 2) and that highest PFe concentration coincide with slightly lower but comparable LPUnFe concentrations in the water column (Fig. 2). This is not surprising (see details below), but we wanted to highlight this finding (section 3.1.1) before going into detail and describe sources and sinks of DFe and PFe. Here the explanation. We assume that particles in unfiltered seawater are identical to the particles collected by SAPS and just an incomplete dissolution of particles in unfiltered seawater should affect a similar result. All seawater samples were acidified with $HNO_3$ to pH 1.7 and stored for 1 year (supp. Text S1 line 36). Our data shows that this long and strong acidification re-dissolved 60% of particulate Fe from associated SAPS samples. The remaining 40% must belong to insoluble silicates. This is stated in the manuscript. The leachable particulate fraction from SAPS particles (LPFe) was determined by a 25% HAc leach for 2 hours at room temperature. It is wrong to believe that the $HNO_3$ and HAc leach are equal and deliver similar results. The pH of a 25% HAc solution is above 2.1 and applied for a short period of time. The HAc leach is usually strong enough to re-dissolve just surface scavenged, some biogenic trace metals and trace metals in carbonates (see any sequential extraction schemes for sediments). As shown in Table 2, on average less than 1% of the Fe belongs to the LP fraction (SAPS), the remaining 99% belong to the refractory fraction (which includes oxides, silicates, ect.). However, with respect to that the $HNO_3$ leach of unfiltered seawater must clearly also re-dissolve particulate trace metal oxides and amorphous substances. Thus comparing LPUn and LP makes no sense. In addition we have not introduced the refractory and the leachable particulate fraction of SAPS particles. We just compare the results from two particulate fractions that showed comparable results! However, we slightly modified

the sentence at line 161ff and replaced "refractory" by "insoluble" in line 171. We agree refractory is misleading since it does not rely on RPFe in SAPS samples.

-For the second point, presumably, they want to use the bottle-derived leachable particulate data because of higher depth resolution than SAPS samples, so this should be a separate argument than the one about the presence of refractory Fe and Al. Answer: Exactly, see explanation above.

-Table 2: are the LPFe, LPAl, LPMn columns derived from the SAPS LP or the bottle LPun? This should be specified in the caption. The units for the LP are specified as being in percent. But these values are all very smallâËŸA ËĞ Tmostly less than 5%, even for Mn, which contradict what was stated in the text in lines 161-163: "The LPUn corresponded to ca. 63_4% of the PFe, 83_11% of the PAl and 100_10% of the PMn fractions." Presumably, this discrepancy arises because the Table 2 values are SAPSbased (LP/P), whereas the values in the text are mixing and matching bottle-based LPun and SAPS-based totals. Is that correct? If this is the case, then this means that the bottle-based LPun are much higher than the SAPS-based LPs. One might expect this to some degree, since the LPun is acidified to a slightly lower pH (_1.7) and for a lot longer than the SAPS-based LPs (25% acetic acid should have a pH _ 2.1), but I wouldn't have expected the difference to be so big. This suggests that the particle population collected by the SAPS may be a subset of the particles accessed by the LPun method, and calls into question whether normalizing or comparing the bottle based LPun by/to the SAPS-based P is appropriate or meaningful. As further detailed below, I suggest that the authors do not mix bottle-based and SAPS-based parameters (i.e. they should not report or interpret LPun/P). Answer: As indicated at the end of the caption of Table 2, data corresponds to particulate data from SAPS. We added "SAPS" to the caption in line 827. With respect to the comparison of P and LPUn (comment of the reviewer). We argue in section 3.1.1 that LPUn is a subset of P, not the other way round. As stated earlier we try in section 3.1.1 not to compare both data sets, because this is to some extend meaningless (we do not know what did not dissolve in unfiltered

seawater), but it is without saying interesting that the both parameters LPUn and P show the same distribution in the water column and highlight the strong concentration gradients observed.

Section 3.1.2: this section needs to be considerably revised to account for the following: -Re. lithogenic source of suspended particles: The extremely good correlations of the PFe to PMn and PFe to PAl concentrations (Fig 3a,3b) do support their contention that the particles have a single lithogenic origin (line 197), since biogeochemical processing of the particles would be likely to affect Mn, Fe, and Al differently, and therefore cause more scatter in the data. However, the slopes of their relationships don't support this. The authors base the conclusion that the suspended particles have a lithogenic origin on the slope of the PFe to PMn data (68 mol Fe/mol Mn), which they say agrees well with a typical crustal ratio (they use 60 mol Fe/mol Mn from Wedepohl, which is close to 57 and 50 mol Fe/mol Mn for UCC and BCC crustal averages, respectively, reported by (Taylor and McLennan, 1995)). However, PMn is not usually a good crustal indicator, since Mn-oxides are frequently a large component of marine particulate Mn. For this reason, PAl or PTi are more frequently used to assess how lithogenic the PFe is. However, their PFe to PAl relationship (slope=1.251 mol Fe to mol Al) far exceeds the slope of typical crust (UCC Fe:Al=0.21; BCC Fe:Al=0.41 (Taylor and McLennan, 1995)). Since the sediment elemental ratios from the southern shelf (Table 3, line 199) are close to these crustal averages, this suggests a fairly large component of the PFe that is in excess of a lithogenic source. One can derive a PMn to PAl ratio from their data=PFe:PAl / PFe:PMn = 0.018 mol Mn/mol Al, which is also greater than typical crustal ratios (UCC Mn:Al=0.0037; BCC Mn:Al=0.0082). So just comparing their slopes to crustal ratios to Al, one would expect there to be a fair amount of nonlithogenic Fe and Mn assuming crustal averages are a reasonable approximation of the local sources (a plausible assumption given that the sediment elemental ratios are close to crustal averages). The conclusion that the suspended particles are primarily lithogenic is therefore not supported by their data. Answer: We do not agree with the reviewer. We argue that the strong correlation between P elements supports the existence of a

single source for suspended particles with lithogenic origin (line 197ff). And then we explain in the two following paragraphs why we do think so! As stated in the text the elemental ratio of P is higher as the elemental ratio of S (sediments) and the earth crust (now stated in line 198). We explain in detail that more biogenic Fe in the suspended particulate pool would lower the Fe/Mn ratio (line 205ff), lower than the sediment ratio of 51.1. This is clearly not the case! From line 208 on, we suggest that DFe was scavenged onto inorganic suspended particle surfaces. This process increases the Fe/Mn and Fe/Al ratio of suspended particles, and that is exactly what we observed. We are irritated that the reviewer did not mention the term scavenging a single time, even though this is the driving force for the elevated elemental ratios. In addition, we know that Mn is not a perfect sediment particle tracer and that titanium would be much better for that. However, the Fe/Mn ratio is a nice tracer to differentiate between biogenic and sediment/lithogenic particles!

-The slopes of Fe:Mn and Fe:Al of the LPun fractions are each about half of those for the total particulate fractions (Fig 3a-d). Comparing the leachable and total slopes, this suggests that particulate Fe is about half as labile as particulate Mn or particulate Al. This is much more consistent with their wholly SAPS-based %leachable data from Table 2 (average %leachable Fe < average %leachable Mn_average %leachable Al), than with their LPun/P from lines 161-163 (Fe<Al<Mn), another reason not to mix the bottle-based (LPun) and SAPS-based (LP, P) parameters. Answer: Figure 3c and 3d show the relationship of the leachable particulate elements of unfiltered seawater. The relationship is not as strongly pronounced as for P but exists, and this is an important finding of this study. We stated earlier in section 3.1.1 that on average 100 % of particulate Mn, 63% of particulate Fe and 83% of the particulate Al can be re-dissolved by the acidification of unfiltered seawater with HNO3. This interplay creates the reduced slopes in Fig. 3c and 3d. However, the relationship of LPUn and P is discussed in section 3.1.1 and not in section 3.1.2, the section the reviewer is discussing right now.

-To rule out data quality issues, the authors should report results from an external standard such as a standard reference material that was run alongside the digestion of the suspended particle leaches/digests. External standards in the supplements were only reported for seawater and for sediments, which used a different digestion procedure than the particles. Was there an external standard measured for the suspended particles to ensure that the digestion method was working accurately? Answer: Because of so many different data sets we forgot to include the crm values for SAPS particle digests. We analysed two crm's, TORT-2 and NIST 1573a. Values are now stated in the supplementary Text S1. A link to this data is provided in the main text body, section 2.2 line 134ff.

Section 3.1.3 -Re. meltwater source of LPunFe: is there a relationship between LPunFe and salinity? A scatter plot of these parameters would be evidence of a meltwater source of LPunFe. Answer: We included a graph in the supplementary material (now Fig. S3) illustrating the relationship between salinity vs. dissolved (DFe) and leachable particulate Fe (LPUnFe).

-Lines 253-256: PFe/PAl in krill fecal pellets (0.48) is much lower than PFe/PAl in suspended particles (1.25âËŸAËĞ TFig 3b), though it is relatively close to the sediment ratios from the southern shelf. This suggests that the suspended particles have an additional source of PFe compared to krill or sediments. Answer: On the other hand, the PFe/PMn ratio is almost the same. Schmidt et al. [2016] showed that at some locations on the shelf the content of lithogenic particles exceeds more than 50% of the krill stomach content (Figure 1H). However, we added a possible explanation for this discrepancy (line 256ff).

Section 3.2.1 -The authors should acknowedge that the sediment core data were taken on the nonproductive, southern side of the island, whereas the water column data were taken from the productive, northern side of the island, so benthic fluxes calculated from porewater profiles could be rather disconnected to the measured DFe. Answer: A sentence was included that refers now to benthic flux differences between the northern and southern side of the island (line 291ff). However, benthic fluxes into bottom waters

are irrelevant for our water column fluxes calculations (see supplementary data).

-The authors calculate a vertical supply of DFe to surface mixed layers by assuming a vertical diffusivity and vertical advection (upwelling) used by De Jong et al. 2012 for the region downstream of the Antarctic Peninsula. But later in section 3.4.2, they discuss a study by Tagliabue et al. 2014 modelling DFe supply for South Georgia, in which Ekman *downwelling* (not upwelling!) prevailed, removing DFe from the surface, rather than supplying it. I can appreciate that there's uncertainty in the estimate of vertical advection, but they should probably pick a sign and stick to it! Answer: Both regions, the shelf and bloom region, are separated by 1.250km and have to face different oceanographic settings. This is stated in the manuscript section 3.4, line 448. Tagliabue et al. 2014 modeled the dissolved Fe supply over the entire Southern Ocean. This model, which is based on observational data, suggests that the main Fe source south of the Polar Front (PF) is Eckman upwelling of nutritious deep waters. North of the Polar Front (South Georgia is located north of the PF) the advective term of vertical missing is negative (downwelling) and the deep winter entrainment is responsible for a large fraction of primary productivity found. However, near an island with shallow topography this scenario is entirely different, and that is true even north of the PF. As already pointed out, internal waves that cross a shallow shelf produce large turbulences, which supports positive vertical advective mixing of nutritious deep waters into the surface layer [Kurapov et al., 2010; Moore, 2000; Wolanski and Delesalle, 1995]. This turbulences are even strong enough to bring sediment particles in re-suspension (what we see on the shelf). Because of that we applied in addition to the diffusive mixing term also the advective term. The Kz and w terms were inferred from de Jong et al. (2012) collected near the Antarctic Peninsula. For our opinion these numbers reflect the advective and diffusive force best for a shallow shelf in the Southern Ocean. However, for clarification we replaced the term upwelling by "advective" velocity and Eckman "transport".

Section 3.3: there are some problems with the proposed flowpaths: -Line 384: I do not understand the proposed advective pathway: the authors reference "Fig. 1; #11/12

via #13 to #14" as a NE-SW transect. First, these stations do not define a NE-SW transect. Second, what is the order of the proposed advective pathway? Is it 13 to 14 to 11/12, even though 13 is actually further from the island than 14? Is 13 shallower than 14 (Fig 4) despite being further offshore (Fig 1)? Answer: We thank the reviewer for their comment. Indeed the pathway is from St. 14 near onshore via St.13 to St. 11/12 offshore. We changed that in the text and mention that the pathway is in E-W direction. Because stations were sampled one after another and are located on an almost linear line from 25 km away from the coast to 70 km offshore we decided to choose this transect. On the other hand this transect represents the only almost linear transect from onshore to offshore with more than two stations, as mentioned later by the reviewer. However, shelf topography of South Georgia is highly variable and varies usually between 200 and 250 m depth (Fig. 2). Nevertheless, locations with a shallower water columns are possible, as we found this for site #13 with ca. 133 m depth. However, this is what we found!

-Presuming that the authors are assuming that #11/12 is the offshore end of the flow-path, this is inconsistent with DMn, which is higher at 11/12 compared to 13 and 14. It's no wonder that they were not able to fit an exponential function to the DFe data (lines 409-410). If this isn't an advective flowpath, the good exponential fit to the LPUnFe data may be a coincidence, or at least unrelated to offshore transport. Further, the "exponential decrease" in PFe was based on a 2 point fit! No wonder $R^2$=1! This should be removed. Answer: We were also surprised that LPUnFe concentrations decreased exponentially with increasing distance to the coast so well. However, suspended sediment particles sink quickly when turbulent mixing decreases. Due to this we expect that shortly after the shelf break was passed most sediment particles are vertically exported. This is in agreement with the literature. So, chances are very slim that these sediment particles below the mixing layer between 200 m and 400 m contribute significantly to the Fe budget approx. 1,250 km downstream the island. Particles mainly sink! We mention this here because the reviewer said earlier we introduced an PFe fraction here that was not taken into account for the later budget. However, to clarify this we

added a sentence (line 420ff). However, this is different for DFe. This Fe fraction can vertically rise, driven by vertical diffusive mixing and strong concentration gradients. Anyway, for a more robust horizontal flux calculation we decided to take the entire DFe data into account. Because of the large variability of the data set and an overall short transect length (max. 80 km away from the coast; de Jong et al. (2012) showed data 3500 km downstream the source region) we could not fit a line through the data, which is needed to calculate the scale length and Kh (horizontal diffusivity (de Jong et al. (2012)). Sorry, this is what we found and I think this is the best way to handle the data carefully.

Section 3.4: budget -As noted above, there is a discrepancy in the manuscript about whether this region is downwelling, as assumed here for the budget, or upwelling, as assumed in section 3.2.1âËŸAËĞ Tsediment DFe flux. Answer: This has been discussed in detail earlier. However, the reviewer must have not understand that the vertical advective mixing is different at the Fe source region and the region of elevated photosynthetic activity [Tagliabue et al., 2014]). However, we included this now in the text section 3.2.1, line 328ff) and leave the discussion open if the advective term should be used or not.

-In the manuscript, an overall vertical loss of DFe of -0.0025 umol DFe/m2/d is assumed (line 465), but in Figure 8, a vertical supply of +0.009 umol DFe/m2/d is indicated. Answer: As pointed out in Figure 8 the vertical input from below is the sum of diffusive (diapycnal = 0.0003 umol m-2 d-1), advective (Eckman = -0.0028 umol m-2 d-1) and deep winter mixing (entrainment = 0.011 umol m-2 d-1). With respect to the fourth and fifth decimal place we calculate a vertical flux from below ca. 0.009 umol m-2 d-1.

-Given the poor (lack of) constraints on both horizontal and vertical supply of DFe, the suggestion that there is a mismatch in the supply and demand that is filled by advecting biogenic Fe is a nice hypothesis, but rather speculative, both the size of the term, and the nature of it. The size of the flux is not constrained, since they did not have local estimates of horizontal or vertical supply. Re. the nature of their proposed flux: their

PFe:PAl ratio from earlier in the manuscript does suggest excess particulate Fe over lithogenic, so this is at least consistent with an available biogenic Fe pool on the shelf, but is also consistent with Fe oxyhydroxides (i.e., authigenic, not biogenic particulate Fe). Answer: Our observational data from the shelf indicates that the South Georgia shelf represent a strong DFe source that supports elevated primary production and the growth of larger organism (located at a higher trophic level) on the shelf. Fe fluxes calculated for the bloom region rely on validated literature values (as this is described in the text extensively). Tagliabue et al. (2016) and Chance et al. [2015] showed that aeolian dust deposition, horizontal and vertical Fe input delivers enough DFe to support phytoplankton growth in most low productivity waters in the Southern Ocean. However, with respect to the estimated Fe requirements of the phytoplankton community in high productivity waters, such as downstream the island of South Georgia, these sources are two low to explain the observed growth. So, DFe needs to come from somewhere! Tagliabue et al. (2016) argued that mainly the entrainment of DFe during the winter drives the phytoplankton growth. However, winter entrainment is a one-time event in the winter and increases the DFe pool in surface waters temporarely. He suggests that most of this DFe is incorporated in phytoplankton cells (biogenic Fe by luxury Fe uptake) in the early time of the growing season. This biogenic is then internally recycled in the surface mixed layer during the entire growing season. We argue that this on-time shot of DFe is not enough in high productivity waters. We suggest that in addition Fe enriched biogenic particles/phytoplankton cells are formed by luxury Fe uptake on the shelf and that these Fe enriched biogenic particles are internally recycled when water masses are transported downstream away from the island. See section 3.4.5 line 516ff.

Miscellaneous: -Lines 104-107: I assume that unfiltered samples were also collected from the tow fish in JR247, since LPun values are plotted for this cruise in figure 6. Please add this to this section. Answer: We added "Unfiltered surface seawater samples were collected and dispensed into acid washed 125 mL LDPE bottles." line 107ff.

-Line 231-2: Fig 6a shows that surface concentrations of 0.3 _M LPunFe are quite

common on the shelf, but 22 uM doesn't show up in this figure, so referring to it here is not that helpful. Answer: We removed the figure citation. However, LPUnFe concentrations are usually lower around 2.2 uM near the shore and <10 nM after the shelf break. We decided to not us a log scaling just for one data point.

Typos: Line 106: "cartridge filter", not "filer" Line 178: "where", not "were" Line 251: "ascent" not "ascend" Line 415: add "there was" to "...that *there was* a limited transfer..." Line 459: "Tagliabue" not "Tagliabu" Answer: Changed as suggested.

References Taylor, S.R., McLennan, S.M., 1995. The Geochemical Evolution of the Continental- Crust. Reviews of Geophysics 33 (2), 241-265. Interactive comment on Biogeosciences Discuss., https://doi.org/10.5194/bg-2017-299, 2017. Chance, R., T. D. Jickells, and A. R. Baker (2015), Atmospheric trace metal concentrations, solubility and deposition fluxes in remote marine air over the south-east Atlantic, Mar. Chem., 177, 45-55. Graham, R. M., A. M. De Boer, E. van Sebille, K. E. Kohfeld, and C. Schlosser (2015), Inferring source regions and supply mechanisms of iron in the Southern Ocean from satellite chlorophyll data, Deep Sea Research Part I: Oceanographic Research Papers, 104, 9-25. Kurapov, A. L., J. S. Allen, and G. D. Egbert (2010), Combined Effects of Wind-Driven Upwelling and Internal Tide on the Continental Shelf, Journal of Physical Oceanography, 40, 737-756. Moore, W. S. (2000), Determining coastal mixing rates using radium isotopes, Cont. Shelf Res., 20(15), 1993-2007. Schmidt, K., C. Schlosser, A. Atkinson, S. Fielding, H. J. Venables, C. M. Waluda, and E. P. Achterberg (2016), Zooplankton gut passage mobilises lithogenic iron for ocean productivity, Curr. Biol., 26, 1-7. Tagliabue, A., J.-B. Sallee, A. R. Bowie, M. Levy, S. Swart, and P. W. Boyd (2014), Surface-water iron supplies in the Southern Ocean sustained by deep winter mixing, Nat Geosci, 7(4), 314-320. Wolanski, E. J., and B. Delesalle (1995), Upwelling by internal waves, Tahiti, French Polynesia, Cont. Shelf Res., 15, 357-368.

---

## Referee Comment (RC2) · Anonymous Referee #2 · 6 Feb 2018

This study proposes a discussion of the sources of iron and related elements (Mn, Al) to the South Georgia region. A major objective of the study is to quantify the sources of iron (dissolved and particulate) on the shelves surrounding the island and to discuss how this iron is transported offshore to sustain the strong bloom that is observed over hundreds of kilometers downstream of the island. This study is based on observations collected during three cruises (2 cruises dedicated to seawater sampling and one cruise during which sediment cores have been collected). This study is rather interesting and provides a nice collection of data very useful to understand and constrain the iron cycle in that important and particular region. In particular, they have estimated the potential impact of grazers (more specifically krill) which is sufficiently rare to be no-

ticed. The author suggests that krill play an important role in the delivery of iron on the shelf that is even more important than diffusive and advective fluxes from the sediments and than meltwater supply (which remains unconstrain). They also state that a large fraction of the iron that is supplied to the region downstream of the island is transported as biogenic particulate iron. I had some major issues concerning this paper. However, most of them have been addressed in the responses to reviewer 1 which are available on the website. Since, for most of them, I think they are appropriate, i focus here my concerns to the unaddressed ones.

I think that the budget is rather speculative and very uncertain. Many numbers are based on a study (De Jong et al., 2012) performed in a different region, that is the Antarctic Peninsula. This area shares some similarities with the South Georgia Island: a shelf area located in the Southern OCean. However, this does not guarantee that the numbers (diffusive and advectives fluxes) are comparable. Many processes may be significantly different such as tidal mixing, tidal residual current, upwelling (or downwelling) over the shelf, inertial waves, ... As a consequence, I would say that the similarity in terms of geography does not necessarily support the idea that processes should be identical. I understand that better constraining the numbers is a very difficult (if not impossible) task. However, uncertainties should be more extensively discussed.

The authors suggest that a large part of the offshore supply from the island to the downstream region is sustained by lateral transport of biogenic materials rich in iron (luxury uptake by phytoplankton on the shelf). That's a valid explanation. However, there are some other potential explanations. For instance, labile iron hydroxydes formed on the shelf can also be transported offshore. Iron adsorbed onto biogenic (or non biogenic) particles can also be advected offshore. This should be also discussed by the authors. Over the shelf, the authors state that a large fraction of the iron is being supplied by krills which ingest lithogenic materials (while filtering seawater) and release it as either dissolved iron or as particulate iron within fecal pellets. However, they assume that all of the excreted and egested iron is available as new dissolved iron. This is

true if this process solubilizes part of the refractory material which would be otherwise unavailable. This seems to be the case since the LPFe fraction is higher in fecal pellets (∼2.5%) than in suspended materials (<1%). This is also true if krills drive a net transport of iron, for instance from the sediment to the open ocean. Again this seems to be true as krills are feeding, at least partly, on sedimentary materials. However, the contribution of this source of food to the total diet of krills is unknown. And thus, the net source of iron due to krills should be uncertain. In other words, it's impossible to quantify the amount of iron that is newly supplied to the system (either by feeding on sediments or by solubilizing an otherwise unavailable iron pool) and the amount of iron that is recycled within the system (grazing on suspended particulate materials and living organisms). This should be better discussed in the manuscript.

Finally, I have a more specific comment already made by reviewer 1 and that has not been really discussed by the authors. They claim that LPFe exhibits an exponential decrease with the distance from the coast. That's a rather strong assumption knowing that the relationship is derived from three points in one case (stations 14, 13, and 11/12) and from 2 points in the other case (14 and 13). I may have misunderstood something (a plot would help) but 3 or 2 points are not enough to constrain the shape of a function (when this shape is unknown). This explains the very high R2. For instance, 2 points could be fitted by a linear function, a polynomial function, or any continuous function ...

In conclusion, I think this paper has the potential to be published in biogeosciences. However, it should be significantly revised before.

---

## Author Comment (AC2) · 27 Feb 2018

We thank reviewer2 for their useful comments and hope to have addressed them appropriately in the following paragraphs.

Reviewer2 stated that the Fe budget is rather speculative and very uncertain. We agree with the reviewer that estimates for Fe budgets are challenging and in most cases contain large uncertainties. This is primarily the result of available flux data that is strongly limited temporally and spatially. Recently, I had a very interesting conversation about flux estimates with Prof Dr. Peter Brand from GEOMAR on research cruise M145 on RV Meteor, which is taking place right now in the tropical Atlantic Ocean. He pointed

out that vertical diffusivity values, for instance, are strongly variable and that mean annual vertical diffusivity estimates, necessary for precise oxygen supply calculations, reached a high level of accuracy only after the same tropical region had been monitored for 12 years with more than 10 research cruises and high resolution CTD, glider and mooring deployments. However, the Southern Ocean is not such a well-investigated region, knowledge gaps are wide and values such as vertical advective and diffusive mixing critical for flux estimates are not well constrained. As correctly pointed out by reviewer2, even if such parameters are constrained for one region it does not mean that they can be used for other areas without introducing large uncertainties. Nevertheless, flux estimates even with large uncertainties can help to understand the degree of the nutrient supply vs. consumption by organisms and help to grasp the limitation of the estimates made. We agree with reviewer2 that our manuscript in the present stage falls to far short in discussing the degree of uncertainty of the Fe budget. Because of that we will add another set of paragraphs below chapter 3.4.5 to discuss the limitation of the Fe budget and the degree of uncertainty.

We agree that we should have provided a more rounded consideration of other potential processes for transporting Fe off the shelf, including non-biological candidate mechanisms. To explain other potential mechanisms we will add another paragraph into section 3.4.5. In relation to the specific mechanisms posed by the second reviewer, we agree that Fe freshly absorbed onto biogenic and non-biogenic material can be released and incorporated by phytoplankton and bacteria. However, the bioavailability of adsorbed and inorganic Fe changes over time! Both Chen and Wang (2001) and Wells et al. (1991) demonstrated that the bioavailability of freshly precipitated Fe oxyhydroxides and Fe adsorbed onto colloids/inorganic particles decreases over time. This is primarily due to the dehydration of the loosely packed structure that is subsequently transferred into amorphous FeOOH in the mineral structure Goethite. Because of this we suggest that the majority of Fe from inorganic FeOOH or Fe adsorbed onto particles must be released and utilized at an early stage of the voyage, mainly on the shelf or shortly after the shelf break. In contrast, suspended inorganic Fe particles or Fe

adsorbed onto particle surfaces farther away are older and became less bioavailable. These issues will be discussed in more detailed in our section 3.4.5.

Reviewer 2 argues that the supply of "new" Fe by krill feeding on sediment particles cannot be quantified. It is true that the experimental set-up did not allow us to specifically distinguish between "recycled" Fe from organic material and "new" Fe from krill grazing on sediments. However, Schmidt et al. (2016) concluded that zooplankton gut passage mobilizes lithogenic Fe, and showed that there are strong spatial patterns in the organic and lithogenic make-up of fecal pellets. This included an exponential decline in the quantity of lithogenic particles in krill stomachs with distance from sources of glacial flour on the northern South Georgia coast. For instance, the lithogenic content at one site on the shelf contributed ∼90% suggesting that a large quantity of the accessible Fe was remobilized from those inorganic particles. However, during the present study we were focusing mainly on net Fe fluxes and fewer on new and recycled Fe. Nevertheless, we are adding more insight into this problem in section 3.2.3., including a numerical exploration of a range of scenarios of the relative proportion of "new" and "recycled" Fe produced by krill

In section 3.3 we discussed the lateral distribution of LPFe and PFe along a NE – SW transect and applied an exponential curve fit to the three data points of LPFe and the 2 data points of PFe. Both reviewers argue that this is not appropriate. We understand that an exponential fit through 2 data points is meaningless. For three data points this approach is not much better. However, when we investigated the data in the early stage of the manuscript we were really surprised to see a very high R2 value for an exponential curve fit of the LPFe fraction. Because of that good R2 we thought it would be a good idea to share that finding with the readership of the article. However, we understand that such curve fittings can be wrongly interpreted and beside that do not contain a strong message for the manuscript. We are therefore removing the exponential equations and deleting the word "exponentially" from line 394.

References

Chen, M. and Wang, W.-X.: Bioavailability of natural colloid-bound iron to marine plankton: Influences of colloidal size and aging, Limnol. Oceanogr., 46, 1956-1967, 2001. Schmidt, K., Schlosser, C., Atkinson, A., Fielding, S., Venables, H. J., Waluda, C. M., and Achterberg, E. P.: Zooplankton gut passage mobilises lithogenic iron for ocean productivity, Curr. Biol., 26, 1-7, 2016. Wells, M. L., Mayer, L. M., Donard, O. F. X., Sierra, M. M. D., and Ackelson, S. G.: The Photolysis Of Colloidal Iron In The Oceans, Nature, 353, 248-250, 1991.

---

## Author Response (AR1)

**Reply to both reviewer**

We would like thank both reviewers for reviewing this manuscript. Both reviewers commented fairly and revealed the weak points of our manuscript. We honestly apologize to reviewer 1 for some exaggerated comments in our first reply. However, we have considered the comments from both reviewers carefully and addressed or fully integrated them in the revised version that we have submitted. In addition, we have made small changes throughout the manuscript (text, tables, and figures (marked yellow)) to make it easier for the reader to follow our interpretations of the different trace metal data sets (two separate particulate fractions, the leachable particulate fraction, and the dissolved trace metal fraction). We have also added a section discussing uncertainties associated with our DFe budget in the bloom region north of South Georgia. We are, for instance, well aware that our Fe budget calculations rely heavily on literature values, and that, in the cases of atmospheric dust deposition and horizontal fluxes, were adapted from another region in the Southern Ocean. It was this approach or no budget at all, given the gaps in observational data for the region.  Our estimates are not unique in this regard. Almost all ocean DFe budgets rely to some extent on such an approach and also suffer from similar large uncertainties that result. This is now discussed in more detail in section 3.4.6 Budget uncertainties.  Furthermore, we included windows comments in the track changed document below to explain made changes and relate them to the reviewers comments.

Methods: -Why were sediment cores collected on the southern
shelf and not the northern shelf? Given that the phytoplankton bloom appears to original
on the northern shelf, and all water column and particle samples were collected
on the northern shelf, this seems like a strange decision. If this was outside of the authors'
control, then this needs to at least be discussed when sediments are compared
to the water column, since one would expect the stronger bloom on the northside to
affect the benthic processing of Fe and Mn, so sediments on the southern shelf may
not be representative of those on the northern shelf.
Answer: Line306ff

-Fecal pellet data are first discussed in section 3.1.3 (for data in Table 4) but without
introducing the methodology for fecal pellet sampling and analysis. They are discussed
again in section 3.2.3, where a reference is to the Schmidt et al papers, but this should
be moved up.
Answer: Line 145ff

Section 3.1.1 -The discussion of particulate fractions in this section is confusing. LPun
is derived from acidified unfiltered seawater and defined in equation 1, but then is
compared to P as if they should be the same, differing only in their sampling methods
(acidified unfiltered seawater vs SAPS sampleâËŸAË˘G Tlines 157-158). However, P is
defined in section 2.2 as the sum of leachable (LP) and refractory (RP) fractions from
the SAPS samples, so one wouldn't expect LPun and P to be the same. I initially read
this section thinking that they were comparing LPun (from acidified unfiltered seawater)
and LP (from acetic acid leach of SAPS samples), which is the more direct comparison
if they want to isolate sampling differences (but still not perfect since acidification to
pH 1.7 with HNO3 is still not the same as a 25% acetic acid leach, but at least more
comparable). This section needs to be clarified. It seems that they have two points in
this section: 1) that pFe and pAl have a refractory component, since LP(un)/P < 100%, and 2) LP(un)
scales with P, so they want to justify using LP(un) for P. For the first point,
it seems that a wholly SAPS-derived assessment of LP/(LP+RP) would be the better
parameter to present, because then there is no confusion of mixing sampling systems,
pore sizes, and leach types. This is done in section 3.2.2 (lines 320-340), which would
make more sense in this section
Answer: Line 179ff and modifications throughout the manuscript

Table 2: are the LPFe, LPAl, LPMn columns derived from the SAPS LP or the bottle
LPun? This should be specified in the caption. The units for the LP are specified
as being in percent. But these values are all very smallâËŸA Ë˘G Tmostly less than
5%, even for Mn, which contradict what was stated in the text in lines 161-163: "The
LPUn corresponded to ca. 63_4% of the PFe, 83_11% of the PAl and 100_10% of
the PMn fractions." Presumably, this discrepancy arises because the Table 2 values
are SAPSbased (LP/P), whereas the values in the text are mixing and matching bottlebased
LPun and SAPS-based totals. Is that correct? If this is the case, then this means
that the bottle-based LPun are much higher than the SAPS-based LPs. One might
expect this to some degree, since the LPun is acidified to a slightly lower pH (_1.7)
and for a lot longer than the SAPS-based LPs (25% acetic acid should have a pH _
2.1), but I wouldn't have expected the difference to be so big. This suggests that the
particle population collected by the SAPS may be a subset of the particles accessed by
the LPun method, and calls into question whether normalizing or comparing the bottle
based LPun by/to the SAPS-based P is appropriate or meaningful. As further detailed
below, I suggest that the authors do not mix bottle-based and SAPS-based parameters
(i.e. they should not report or interpret LPun/P).
Answer: See Table 2

-Re. lithogenic source of suspended particles: The extremely good correlations of the
PFe to PMn and PFe to PAl concentrations (Fig 3a,3b) do support their contention that
the particles have a single lithogenic origin (line 197), since biogeochemical processing
of the particles would be likely to affect Mn, Fe, and Al differently, and therefore cause
more scatter in the data. However, the slopes of their relationships don't support this.
The authors base the conclusion that the suspended particles have a lithogenic origin
on the slope of the PFe to PMn data (68 mol Fe/mol Mn), which they say agrees
well with a typical crustal ratio (they use 60 mol Fe/mol Mn from Wedepohl, which is
close to 57 and 50 mol Fe/mol Mn for UCC and BCC crustal averages, respectively,
reported by (Taylor and McLennan, 1995)). However, PMn is not usually a good crustal
indicator, since Mn-oxides are frequently a large component of marine particulate Mn.
For this reason, PAl or PTi are more frequently used to assess how lithogenic the PFe
is. However, their PFe to PAl relationship (slope=1.251 mol Fe to mol Al) far exceeds
the slope of typical crust (UCC Fe:Al=0.21; BCC Fe:Al=0.41 (Taylor and McLennan,
1995)). Since the sediment elemental ratios from the southern shelf (Table 3, line
199) are close to these crustal averages, this suggests a fairly large component of the
PFe that is in excess of a lithogenic source. One can derive a PMn to PAl ratio from
their data=PFe:PAl / PFe:PMn = 0.018 mol Mn/mol Al, which is also greater than typical
crustal ratios (UCC Mn:Al=0.0037; BCC Mn:Al=0.0082). So just comparing their slopes
to crustal ratios to Al, one would expect there to be a fair amount of nonlithogenic
Fe and Mn assuming crustal averages are a reasonable approximation of the local
sources (a plausible assumption given that the sediment elemental ratios are close to
crustal averages). The conclusion that the suspended particles are primarily lithogenic
is therefore not supported by their data.
Answer: Line 214ff

-To rule out data quality issues, the authors should report results from an external standard such as a
standard reference material that was run alongside the digestion of the
suspended particle leaches/digests. External standards in the supplements were only
reported for seawater and for sediments, which used a different digestion procedure
than the particles. Was there an external standard measured for the suspended particles
to ensure that the digestion method was working accurately?
Answer: See Text S1

Section 3.1.3 -Re. meltwater source of LPunFe: is there a relationship between
LPunFe and salinity? A scatter plot of these parameters would be evidence of a meltwater
source of LPunFe.
Answer: See Fig. S1

-The authors calculate a vertical supply of DFe to surface mixed layers by assuming a
vertical diffusivity and vertical advection (upwelling) used by De Jong et al. 2012 for the
region downstream of the Antarctic Peninsula. But later in section 3.4.2, they discuss
a study by Tagliabue et al. 2014 modelling DFe supply for South Georgia, in which Ekman
*downwelling* (not upwelling!) prevailed, removing DFe from the surface, rather
than supplying it. I can appreciate that there's uncertainty in the estimate of vertical advection,
but they should probably pick a sign and stick to it!
Answer: Line 334ff

Section 3.3: there are some problems with the proposed flowpaths: -Line 384: I do
not understand the proposed advective pathway: the authors reference "Fig. 1; #11/12 via #13 to #14"
as a NE-SW transect. First, these stations do not define a NE-SW
transect. Second, what is the order of the proposed advective pathway? Is it 13 to 14
to 11/12, even though 13 is actually further from the island than 14? Is 13 shallower
than 14 (Fig 4) despite being further offshore (Fig 1)?

Answer: Line 421ff
-Presuming that the authors are assuming that #11/12 is the offshore end of the flowpath,
this is inconsistent with DMn, which is higher at 11/12 compared to 13 and 14. It's
no wonder that they were not able to fit an exponential function to the DFe data (lines
409-410). If this isn't an advective flowpath, the good exponential fit to the LPUnFe
data may be a coincidence, or at least unrelated to offshore transport. Further, the "exponential
decrease" in PFe was based on a 2 point fit! No wonder R2=1! This should
be removed.
Answer: Line 432ff
-In the manuscript, an overall vertical loss of DFe of -0.0025 umol DFe/m2/d is assumed
(line 465), but in Figure 8, a vertical supply of +0.009 umol DFe/m2/d is indicated.
Answer: See Fig.8
-Given the poor (lack of) constraints on both horizontal and vertical supply of DFe, the
suggestion that there is a mismatch in the supply and demand that is filled by advecting
biogenic Fe is a nice hypothesis, but rather speculative, both the size of the term, and
the nature of it. The size of the flux is not constrained, since they did not have local
estimates of horizontal or vertical supply. Re. the nature of their proposed flux: their PFe:PAl ratio
from earlier in the manuscript does suggest excess particulate Fe over
lithogenic, so this is at least consistent with an available biogenic Fe pool on the shelf,
but is also consistent with Fe oxyhydroxides (i.e., authigenic, not biogenic particulate
Fe).
Answer: See Abstract, Conclusion, Section 3.4.6 and Line 474ff

**Reviewer2**
I think that the budget is rather speculative and very uncertain. Many numbers are
based on a study (De Jong et al., 2012) performed in a different region, that is the
Antarctic Peninsula. This area shares some similarities with the South Georgia Island:
a shelf area located in the Southern OCean. However, this does not guarantee
that the numbers (diffusive and advectives fluxes) are comparable. Many processes
may be significantly different such as tidal mixing, tidal residual current, upwelling (or
downwelling) over the shelf, inertial waves, ... As a consequence, I would say that the
similarity in terms of geography does not necessarily support the idea that processes
should be identical. I understand that better constraining the numbers is a very difficult
(if not impossible) task. However, uncertainties should be more extensively discussed.
Answer: See section 3.4.6
The authors suggest that a large part of the offshore supply from the island to the downstream
region is sustained by lateral transport of biogenic materials rich in iron (luxury
uptake by phytoplankton on the shelf). That's a valid explanation. However, there are
some other potential explanations. For instance, labile iron hydroxydes formed on the
shelf can also be transported offshore. Iron adsorbed onto biogenic (or non biogenic)
particles can also be advected offshore. This should be also discussed by the authors.
Over the shelf, the authors state that a large fraction of the iron is being supplied by
krills which ingest lithogenic materials (while filtering seawater) and release it as either
dissolved iron or as particulate iron within fecal pellets. However, they assume
that all of the excreted and egested iron is available as new dissolved iron. This is true if this process
solubilizes part of the refractory material which would be otherwise
unavailable. This seems to be the case since the LPFe fraction is higher in fecal pellets
(_2.5%) than in suspended materials (<1%). This is also true if krills drive a net
transport of iron, for instance from the sediment to the open ocean. Again this seems

[revised manuscript text omitted]

**Kommentar [SC2]:** As suggested by reviewer1 we included the following sentence.

**Kommentar [SC3]:** We included now why sediment samples were sampled on the southern side of the shelf

Krill faecal pellets were obtained during on-board krill incubations performed during JR247.

Incubations were performed in darkness in the laminar flow cabinet at ambient surface layer temperature. The krill were incubated in filtered seawater from the tow fish for up to ~3 h immediately after capture, so that pellets obtained derived from material ingested in situ. These incubations and their results are described in more detail in Schmidt et al. (2016).

**2.2 Trace metal analysis in suspended particles and krill faecal pellets**

The labile trace metal fraction of suspended particles (SAPS) and krill faecal pellets, was remobilized using a 25% acetic acid solution (glacial SpA, Romil) following Planquette et al. (2011). The labile trace metal fraction is hereafter referred to as the leachable particulate trace metal fraction (LP). The remaining particles were digested on a hot plate applying a mixture of aqua regia and hydrogen fluoride (Planquette et al., 2011). This fraction will be referred to as the refractory particulate fraction (RP). The particulate trace metal fraction (P) is the sum of leachable particulate (LP) and refractory particulate (RP). All samples were analysed by collision cell inductively coupled plasma - mass spectrometry (ICP-MS) (ThermoFisher Scientific, XSeriesII). For more detailed description of measured certified reference material see Supplementary Text S1.

**2.3 Trace metal analysis of seawater**

The filtered and unfiltered seawater samples were stored for a period of 12 months prior to analysis. Concentrations of dissolved and total dissolvable Fe, Mn, and Al in seawater were determined by off-line pre-concentration and isotope dilution / standard addition ICP-MS

(ThermoFisher Scientific Element2 XR) according to Rapp et al. (2017). For a more detailed description of the method and measured reference materials see Supplementary Text S1.

> **Kommentar [SC4]:** We included, how faecal pellets were collected.

**2.4 Trace metal analysis of pore waters and sediments**

Sub-samples of the bulk, homogenized sediments were fully dissolved following an aqua regia and combined hydrofluoric/perchloric acid digestion method following Homoky et al. (2011). The acid digests and pore waters were analysed by ICP-optical emission spectrometry (OES) (Perkin Elmer Optima 4300DV). For a more detailed description of the method and measured reference materials see Supplementary Text S2.

**3. Results & Discussion**

**3.1 Supply routes of suspended particulate Fe, Mn, and Al**

3.1.1 Characterization of (the two) particulate trace metal fractions

**Kommentar [SC5]:** We modified the following section as suggested by reviewer1.

Two different particulate fractions were obtained from samples collected during JR247; a particulate fraction from suspended particles collected using 1 µm pore size SAPS filters (P) and a leachable particulate fraction from unfiltered acidified seawater samples ($LP_{Un}$) collected at the same depth. $LP_{Un}$ was calculated following Eq. (1):

$$LP_{Un} = \text{total dissolvable (TD; unfiltered)} - \text{dissolved (D; 0.2 µm filtered)} \qquad (1)$$

Because of the different sampling approaches (SAPS vs. OTE water samplers), filter sizes (>1 µm for SAPS vs. >0.2 µm for dissolved seawater) and digestion procedures (aqua regia + HF for SAPS particles vs. water sample storage at pH 1.7 [22 µmol $H^+$ $L^{-1}$ ]), concentrations of $LP_{Un}$ and P differed, but showed similar distribution patterns in the water column (Fig. 2, Table 1 and 2). The concentrations of Fe, Mn and Al in the $LP_{Un}$ fraction ($LP_{Un}Fe$, $LP_{Un}Mn$, $LP_{Un}Al$) were slightly lower than the particulate fraction from suspended particles (PFe, PMn, PAl). The $LP_{Un}$ of unfiltered seawater samples corresponded to ~ $63 \pm 4$ % of the PFe, $83 \pm 11\%$ of the PAl and $100 \pm 10\%$ of the PMn fractions obtained by SAPS. The average $LP_{Un}$ trace metal ratios ($LP_{Un}Fe$/ $LP_{Un}Mn = 33.07 \pm$

3.45 (1 σ) and $LP_{Un}Fe/ LP_{Un}Al = 0.65 \pm 0.10$ (n=69)), were about half of the elemental ratios of suspended particles obtained by SAPS ($PFe/PMn = 68.0 \pm 0.6$ and $PFe/PAl = 1.251 \pm 0.042$ (n=42)

(Fig. 3; Table 1 and 2)).

The lower concentrations of Fe and Al and the reduced elemental ratios in the $LP_{Un}$ compared to the P fractions suggests that an unknown fraction of particulate Fe and Al in seawater was not leached during the acidification procedure at pH 1.7 over 12 months. However, since P and $LP_{Un}$

displayed similar trends with depth (Fig. 2), $LP_{Un}$ was used in sections 3.1.3 and 3.3 as an indicator for the abundance of particulate trace metals at locations where particulate samples could not be retrieved by SAPS, e.g. in surface waters collected by the tow fish and depths greater than 150 m.

3.1.2 Suspended particulate trace metals in the water column

Concentrations of PFe, PMn and PAl in the water column ranged from $0.87 - 267$ nmol L$^{-1}$,

$0.01 - 3.85$ nmol L$^{-1}$, and $0.60 - 195$ nmol L$^{-1}$, respectively (Fig. 2, Table 2). Concentrations of

$LP_{Un}Fe$, $LP_{Un}Mn$ and $LP_{Un}Al$ ranged from $1 - 118$ nmol L$^{-1}$, $0.01 - 100$ nmol L$^{-1}$, and $1 - 141$ nmol L$^{-1}$

, respectively (Fig. 2, Table 1). Below the isopycnal density layer 27.05 kg m$^{-3}$, located at ~50 – 70

m depth, P and $LP_{Un}$ increased with depth and showed a maximum near the seafloor of e.g. 207 nmol

L$^{-1}$ for PFe and 112 nmol L$^{-1}$ for $LP_{Un}Fe$ (#17, Table 2). Most sites on the shelf (bottom depth ≤ 260

m; #9/10, #13, #14, #17, and #21) showed seafloor maxima, in agreement with other shelf studies.

For example, Milne et al. (2017) reported concentrations of up to 140 nmol L$^{-1}$ for PFe and 800 nmol

L$^{-1}$ for PAl in bottom waters on the west African shelf, and Chase et al. (2005) showed bottom water maxima of up to 400 nmol L$^{-1}$ for $LP_{Un}Fe$ off the Oregon coast.

Strong linear relationships between elements were observed for suspended particles (SAPS)

obtained from above and below the 27.05 kg m$^{-3}$ isopycnal, with elemental ratios of $PFe/PMn = 68.0$

$\pm 0.6$ and $PFe/PAl = 1.25 \pm 0.04$ (n=42) (Fig. 3, Table 2). These elemental ratios were higher than those reported for the earth's crust (Fe/Mn = 58, Fe/Al = 0.2, Fe/Ti (titanium) = 9.1 (Wedepohl,

1995)) and sediment samples collected to the south of the island (mean sediment surface layer of S1,

**Kommentar [SC6]:** We removed the interpretation of LPUn and P as suggested by reviewer1.

**Kommentar [SC7]:** For clarity, we indicate now where the LPUn fraction was only applied

**Kommentar [SC8]:** We changed the paragraph as suggested by reviewer 1.

S2, S3; SFe/SMn = 51.5 ± 2.4, SFe/SAl = 0.34 ± 0.02 (Fig. 4, Table 3), and SFe/STi = 9.9 (not shown)), suggesting that the suspended particles were more enriched in Fe than lithogenic particles.

We are aware that other trace metals, such as Ti, would be more appropriate than Mn to indicate the lithogenic origin of suspended particles. However, the element Ti was not monitored for dissolved, unfiltered seawater and particulate samples obtained by SAPS.

[revised manuscript text omitted]

27) (Table 4) (Schmidt et al. 2016). The molar ratio PFe/PMn = $70.5 \pm 8.21$ of the faecal pellets was similar to those for suspended particles in the water column (PFe/PMn = $68.0 \pm 0.6$; Table 1, 2 and 4), indicating that Fe in krill faecal pellets was predominately associated with terrigenous material and/or glacial flour particles, as also reported by Schmidt et al. (2016). In contrast, the molar ratio PFe/PAl

= $0.48 \pm 0.07$ of faecal pellets was lower than that of suspended particles, PFe/PAl = $1.25 \pm 0.04$, but higher than that of sediments, PFe/PAl = $0.34 \pm 0.02$. The observed variability in the PFe/PAl ratio in the various particle pools is therefore a consequence of different contributions of biogenic material to the particulate reservoir and different amounts of Fe scavenged onto particle surfaces.

**3.2 Supply routes of dissolved Fe, Mn, and Al**

[revised manuscript text omitted]

**Kommentar [SC15]:** We removed the exponential equations and rewrote the sentences.

**Kommentar [SC16]:** We included why particulate trace metals are not included in section 3.4

**Kommentar [SC17]:** As requested by reviewer 1 we included the following sentence.

[revised manuscript text omitted]

Kommentar [SC20]: As requested by reviewer 2, we included alternative explanations.

**3.4.6 Budget uncertainties**

**Kommentar [SC21]:** As requested by reviewer2, we included a section that highlights the uncertainties of our DFe budget.

Estimates for Fe budgets are challenging and often contain large uncertainties. This is primarily due to the lack of site- and time-specific flux data. Moreover, the mean annual estimates, necessary for reliable supply calculations, reach a high level of accuracy only after the same region has been monitored multiple times to cover seasonal and annual anomalies. In the following, we will discuss the uncertainty of the different Fe fluxes in the blooming region north of South Georgia.

We identified three main processes that account together for ~98% of the total Fe flux in the blooming region, and thus contribute largest uncertainties; the horizontal flux, dry/wet deposition, and winter entrainment. Horizontal flux estimates of this study rely on literature values that were collected offshore the Antarctic Peninsula. In contrast, South Georgia is an island with a confined shelf region and thus horizontal DFe fluxes may differ greatly. Furthermore, we showed that dry deposition dust fluxes are generally low, but showed in addition that the Fe flux can be supplemented strongly by sporadic wet deposition events (~ $1.0 \pm 1.2$ μmol DFe m$^{-2}$ d$^{-1}$) (Chance et al., 2015). Atmospheric fluxes are variable, illustrated by the large standard deviation of the wet deposition Fe fluxes obtained at 40°S. Furthermore, to determine the magnitude of the seasonal DFe winter entrainment reliable estimates of the winter mixing layer depth (WMLD) and the ferrocline are required. Even though the WMLD can be estimated very precisely using Argo float data, the depth of ferrocline in the manuscript of Tagliabue et al. (2014) is based on 140 unique observations distributed over the entire Southern Ocean. Due to this regional anomalies are not captured. In addition to the DFe fluxes in the blooming region, we also assume that the biological Fe demand estimated for the phytoplankton community contributes a large error. The biological Fe requirements were determined using satellite derived net primary production data and an average intracellular Fe:C ratio derived from 5 different diatom species native to the Southern Ocean. Both parameters are not well constrained and because of the lack of observational data we applied the lowest intracellular Fe:C ratio available in the literature (Strzepek et al., 2011). However, we found that even small changes of the both parameters change the estimated Fe availability in the bloom region strongly. Nevertheless, flux estimates even with large uncertainties can help us understand the degree of the nutrient supply vs. consumption by organisms and help to pinpoint the limitation of the estimates made. To ultimately reduce the level of uncertainty
and to improve our biogeochemical models more observational data from the bloom region north of
South Georgia is required.

**4. Conclusions**

Shelf sediment-derived Fe and Fe released from Antarctic krill significantly contribute to the
DFe distribution in the shelf waters around South Georgia. Nevertheless, DFe enriched in shelf waters
are not effectively advected to the phytoplankton bloom region downstream of the island. Together
with other Fe supplies, such as aeolian dust, deep winter mixing and diapycnal mixing, the horizontal
advection contributes only ~30% to the Fe requirements of a phytoplankton bloom downstream of
South Georgia. We therefore hypothesize that the majority of the Fe is derived from remineralisation
of Fe enriched phytoplankton cells and biogenic particles that are transported with the water masses
into the bloom region.

> **Kommentar [SC22]:** As requested by reviewer1 we hypothesize now that DFe was in addition transported as biological labile particulate Fe

[revised manuscript text omitted]

Certified reference materials (crm), NIST 1573a and Tort 2, were digested and analysed with each batch of suspended particle and faecal pellet samples, in order to validate our sample concentration. Values obtained agreed with reported values of the crm (NIST 1573a: $423 \pm 5$ mg Fe $kg^{-1}$ (certified $368 \pm 7$ mg Fe $kg^{-1}$), $244 \pm 2$ mg Mn $kg^{-1}$ (certified $246 \pm 8$ mg Mn $kg^{-1}$), $550 \pm 1$ mg Al $kg^{-1}$ (certified $598 \pm 12$ mg Al $kg^{-1}$); Tort-2: $117 \pm 2$ mg Fe $kg^{-1}$ (certified $105 \pm 13$ mg Fe $kg^{-1}$), $13 \pm 1$ mg Mn $kg^{-1}$ (certified $14 \pm 1$ mg Mn $kg^{-1}$)).

**Kommentar [SC27]:** As requested by reviewer1, we included results from analysed crms

[revised manuscript text omitted]

**Kommentar [SC28]:** As requested by reviewer1, we included a figure showing DFe and LPUnFe concentrations in surface waters vs. Salinity

**Figure S1: Tow-Fish surface samples:** Relationship of salinity vs. dissolved (DFe) and leachable particulate Fe (LP$_{Un}$Fe) in surface waters. The Fe concentration along the y-axis is represented in a logarithmic scale. We applied a linear regression, to validate the relationship between the DFe, LP$_{Un}$Fe and salinity (not shown). With exception of the low salinity data point at 33.25 psu, the DFe and LP$_{Un}$Fe vs. salinity data achieved an $R^2$ of 0.46 and 0.38, respectively.

[Figure]

**Figure S2: SAPS samples:** Relationship between leachable particulate (LP) and refractory particulate (RP) Fe, Mn, and Al. Due to the high proportion of RP (98.9 – 99.2% for Fe) in the particulate fraction, using the particulate fraction, P, instead of RP changes the linear regression with LP just very little.

**Kommentar [SC29]:** We changed the order of Figure S1, S2 and S3

[Figure]

**Figure S3: SAPS samples :** Relationship between leachable particulate Fe, Mn and Al.

[Figure]

**Figure S4: OTE-water sampler:** Average dissolved Fe concentration between 100 and 400 m water depth versus distance to the coast line of South Georgia in kilometre.

---

## Author Response (AR2)

*We would like to thank the reviewer for his useful comments and suggestions. We agree with the reviewer that a concrete terminology is required to deliver information in a scientific manuscript and to describe the biogeochemical cycle of compounds in the marine environment. This is sometimes difficult when a large environmental data set with similar parameters is presented and when different scientific fields use the same terminology for processes or species that are different. However, with this revised version we hope to have eliminated all subjects of concern and confusion. For example, we reduced the terminology of particles to biogenic, authigenic and lithogenic (removed terrigenous), describe each terms applied early in the manuscript and used them throughout the manuscript. Furthermore, we included a new table showing the different elemental ratios of the different particle sources.*
*Please find our comments below in italics. Below our comments, we included the revised manuscript and the supplementary material with word track changes.*

For example, the authors seem to use "lithogenic" or "terrigenous" to indicate poorly exchangeable abiotic mineral particles in general, because they are initially interested in distinguishing these types of particles from biogenic particles, whereas I use "lithogenic" and "terrigenous" to specifically mean weathered aluminosilicate material that have elemental ratios consistent with crustal averages, and further separate the general class of abiotic mineral particles into "lithogenic" (i.e. aluminosilicates) vs "authigenic and scavenged" classes based on elemental ratios of sample particles compared to crustal averages. There are, of course, terrigenous and lithogenic particles that have elemental ratios different than crustal averages, so it's not that my version is necessarily better, but I think that if the authors explicitly and clearly define their use and meaning of the words "lithogenic"/ "terrigenous" early in the manuscript, this will help to reduce confusion amongst the general readership about what they are talking about.
*Comment: We agree with the reviewer and removed the term "terrigenious" from the manuscript and replaced it with "lithogenic".*

Another source of confusion was the use of the word "scavenged". This could refer to surface adsorbed Fe, which would probably be readily exchangeable (i.e. bioavailable), but could also refer to authigenic precipitates ranging from rather labile oxyhydroxides that would be accessible by a weak leach to more aged hydroxides and oxides that are not. At various points in the manuscript, the higher Fe/Al ratios they see in suspended particles are referred to as scavenged Fe (e.g. section 3.1.2 lines 412-418) or lithogenic Fe (e.g. section 3.2.2 lines 548-550), depending on the way they are talking about these terms in each of these sections. They should pick a definition of "scavenged" and "lithogenic" that can be used consistently throughout the manuscript.
*Comment: We inserted a new paragraph at the beginning of the RD section and introduce the term biogenic, authigenic (incl. scavenged), and lithogenic particle. This terminology was applied throughout the manuscript.*

I also think it would help if the authors could come up with terminology to more clearly differentiate leachable particulate metals from their 25% HAc leach of their SAPS samples (LP) compared to the leachable particle metals from their total dissolvable seawater samples (LPun). They do define this in section 3.1.1, but in referring to these fractions in the text, they are both sometimes referred to as "leachable" particulate X, even though the former leach is poorly correlated with total particulates, but the latter leach is well correlated and used as a stand-in for total particulates. A more differentiated set of terminology to refer to each of these will further help to clarify.

*Comment: We removed the "P" from "LP" and "RP", and we eliminated the term "particulate" throughout the text that referred to the leachable fraction of particles collected by SAPS. The term "leachable particulate fraction" referrers now to LPUn (unfiltered seawater), while "leachable fraction" and "refractory fraction" referrers to L and R (SAPS).*

Below I highlight specific places where I think additional clarifications will help.

Figure 2 caption: recommend replacing "corresponds to the concentration lables of LPunFe" with "corresponds to the same axis as LPunFe" to clarify the meaning.
*Comment: Changed as suggested.*

Section 3.1.1
Lines 370-372 and Figure 2: The authors state that LPun concentrations "were slightly lower than the particulate fraction", but there are a few instances when LPun Fe exceeds PFe (e.g. deep stn 13, surface stn 18). As the authors note, these are different sampling approaches and pore sizes, so it's not unexpected, but the authors should acknowledge this with language to indicate that the LPun were *usually* lower than P.
*Comment: We included "usually".*

Lines 380-383: Ok, I understand now what the authors are doing by using LPun as an *indicator* of PFe. I transcribed the data from the tables and made a scatter plot of LPun Fe vs PFe and fit a regression to convince myself that they are indeed related (and they are), since it's not always obvious from looking at profiles, but this is a plot that they should include in the supplementals, reporting slope and R2, to justify the usage of LPun to indicate PFe.
*Comment: We included a figure in the supplements showing LPUn vs. P for Fe, Mn, and Al (now Figure S1).*

Lines 396-418: the clarifications added by the authors in the revised version help. I think these paragraphs could be made even clearer by stating at the outset that there are three potential origins to particles: lithogenic, biogenic, and authigenic (including scavenged), and then proceed to argue why the ratios indicate the importance of authigenic precipitation (or scavenging). Note that this is partly a matter of terminology, but what the authors call scavenged, or surface adsorbed, Fe and Mn may not be surface adsorbed to lithogenic or other particles in the traditional sense, but rather exist as discrete Fe and Mn oxyhydroxide mineral particles formed by authigenic precipitation.
I think it would be helpful in this and other discussions if the authors made a table summarizing the Fe/Mn, Fe/Al, and Mn/Al ratios for the particle pools that they talk about (crustal, sediments, suspended particles, fecal pellets, and phytoplankton).
Fe/Mn Fe/Al Mn/Al
crustal 58.00 0.20 0.00345
sediments 51.50 0.34 0.00660
suspended 68.00 1.25 0.01838
fecal pellet 70.50 0.48 0.00681
phytoplankton 1.70
I don't think the additions regarding titanium are necessary (lines 403-405), since the authors already have Al data, which is an adequate lithogenic tracer. I would, however, add a discussion of the ratio of

PMn/PAl as it relates to crustal averages (which shows that PMn is enriched relative to PAl in suspended particles compared to both sediments and the typical crustal average), as this illuminates what is going on with Mn too and gives context for what it means to normalize Fe to Mn, which I still don't really understand.

*We included a new paragraph at the beginning of the RD section introducing the used terminology, biogenic, lithogenic, and authigenic. We added the process of surface scavenging to the authigenic term and changed that throughout the text. In addition we included a new Table (Table2) showing here the different ratios of the different particle pools. We removed Ti and included the PMn/PAl ratio in the text. Here now the explanation, why we are applying the Fe/Mn ratio. Usually we use the Fe/Mn ratio to differentiate between inorganic and organic Fe in marine particles. Both trace metals are incorporated in phytoplankton and bacteria cells. Phytoplankton Fe requirements are high, while Mn requirements are low, thus the intracellular Fe content is higher and the Fe/Mn ratio of phytoplankton cells is low (1.7). Such a low ratio we usually measure for the particulate fraction in open ocean surface seawater, thus we assume that the majority of Fe is part of biogenic fraction, intra- or extracellular. On the shelf, the load of lithogenic particles is much higher (more Mn), with the consequence that the Fe/Mn ratio is higher. This is described in chapter 3.1.2.*

*However, main objective of the paper is to highlight the Fe cycle in the dissolved and particulate fraction and to distinguish between organic and inorganic particulate Fe using the Fe/Al and Fe/Mn ratio as tracers. However, the Fe/Mn ratio of particles is altered by scavenging of DFe and DMn onto their surfaces. Assuming that both dissolved components are scavenged at the same time, MnO is re-dissolved by the process of photoreduction, thus reducing the amount of scavenged Mn (amount unknown) and increasing the Fe/Mn ratio with time from the start point 51.1 (sediment). However, extending the discussion why the Mn/Al ratio is different in the different inorganic particle fractions and thus highlighting the marine Mn cycle (photoreduction, scavenging) would for our opinion distract from the Fe story and did not deliver any new insight to the differentiation of organic and inorganic Fe particles.*

I am also still puzzled as to why the correlations are so good between PFe, PMn, and PAl (Figure 3). I would not expect scavenging (oxidation, precipitation, hydrolysis, etc.) behavior to be the same for these three elements, so why do these correlate so well?

*Commwent: Yes, the correlations of PFe, PMn, and PAl are very good. We were surprised as well when we saw that tight correlation for the first time. However, we showed with the HAc leaches that the majority of PFe, PMn, and PAl were refractory, meaning locked inside the crystal structure of the particle/mineral. We assume that processes such as oxidation, precipitation and hydrolysis did not affect the unaltered composition of the interior of sediment particles. And even when there is a layer of surface scavenged Fe that underwent the proposed processes, their overall contribution to total Fe content (e.g. alumosilicates, in equilibrium with Mn and Al) is very small (see particle leaches).*

Section 3.1.3
Line 429: I suggest saying "enriched with surface bound and authigenic Fe"
*Comment: We included " and authigenic".*

Lines 457-464: The authors conclude that variability in the PFe/PAl ratio in various pools is explained by different contributions of biogenic material and Fe scavenged onto particles surfaces. The biogenic contribution is likely negligible, since Fe concentrations in phytoplankton is so much lower than in lithogenic and authigenic particles (lines 408-409). I agree that variations in the amount of scavenged and authigenic Fe is important for the PFe/PAl ratio. Note that there are large variations in the PMn/PAl ratio too between the different particle pools (see table above), indicating that there is also variation in the scavenged/authigenic Mn, which makes interpreting the Fe/Mn difficult.

*Comment: We agree with the reviewer and changed the sentence to "The observed variability in the PFe/PAl ratio in the various particle pools is therefore a consequence of different amounts of lithogenic and authigenic particles". Difficulties with the Fe/Mn ratio, see comments above.*

The authors conclude that similar PFe/PMn in faecal pellets compared to suspended particles in the water column indicate that "Fe in krill faecal pellets was predominately associated with terrigenous material". Firstly, do they mean terrigenous as opposed to biogenic? If so, which terrigenous? From the molar ratios for all the pools (see table above), the ratios in faecal pellets seem most similar to sediments rather than to suspended particles, so it may be worth highlighting the similarity to sediments instead of comparing to suspended particles.

*Comment: We agree with the reviewer that when taking the Mn/Al and the Fe/Al ratio into account mainly sediments were incorporated in faecal pellets. However, the Fe/Mn ratio suggests that suspended particles were incorporated in faecal pellets. Anyway, we turned the conclusion around, rewrote the sentences, and concluded that different amounts of lithogenic and authigenic particles are responsible for the variability of the elemental ratios.*

Section 3.2.2

Lines 548-550: The authors conclude that the particulate trace metals were "mainly incorporated in lithogenic material", since the leachable fraction from a 25% HAc leach was so small. What do the authors mean here by "lithogenic material", since elsewhere in the text, the authors argue that scavenging of DFe is responsible for increasing the Fe/Al and Fe/Mn in suspended particles compared to sediment and crustal averages (lines 412-418), and that scavenged Fe is exchangeable with DFe in the water column (lines 559-565). 25%HAc is a fairly weak leach, and there can still be authigenic minerals (hydroxides and oxides) that are not accessed by this leach, so do they mean that lithogenic material includes some of these hydroxides and oxides, or do they mean that the suspended particles have a lithogenic (aluminosilicate?) origin that has a much higher Fe/Al and Fe/Mn than the sediments and crustal averages? This point, though seemingly picky, is an example of how it would help to define their use of "lithogenic" more precisely.

*Comment: We agree with the reviewer that "lithogenic" can mean a lot in this sense. However, we meant rock type minerals and replaced "lithogenic" by "mineral structures unaffected by a weak acid leach (e.g. aged oxyhydroxides and alumosilicates)".*

Section 3.2.3

Lines 574-583: the description of the faecal pellet Fe flux is clear, but it would be useful to add in total particulate Fe excreted by krill (faecal pellet mass and % of total Fe that is leachable, so giving an indication of total Fe would complete the picture).

*Comment: Table 5 shows the faecal pellet mass, how much total Fe is incorporated, and how much Fe is leachable in %. These results are based on krill incubation experiments. However, we included in the text, how much Fe (in μmol mg-1) was incorporated in faecal pellets and how much Fe (in nmol mg-1) on average is leachable.*

Typos, wording, etc.

[revised manuscript text omitted]

Certified reference materials (crm), NIST 1573a and Tort 2, were digested and analysed with each batch of suspended particle and faecal pellet samples, in order to validate our sample concentration. Values obtained agreed with reported values of the crm (NIST

1573a: $423 \pm 5$ mg Fe kg$^{-1}$ (certified $368 \pm 7$ mg Fe kg$^{-1}$), $244 \pm 2$ mg Mn kg$^{-1}$ (certified 246

$\pm 8$ mg Mn kg$^{-1}$), $550 \pm 1$ mg Al kg$^{-1}$ (certified $598 \pm 12$ mg Al kg$^{-1}$); Tort-2: $117 \pm 2$ mg Fe kg$^{-1}$ (certified $105 \pm 13$ mg Fe kg$^{-1}$), $13 \pm 1$ mg Mn kg$^{-1}$ (certified $14 \pm 1$ mg Mn kg$^{-1}$)).

[revised manuscript text omitted]

SAPS samplers (P) vs. leachable particulate trace metals from OTE water samplers ($LP_{Un}$).

Data represents the entire data set collected at 20m, 50, and 100/150m.

[Figure]

**Figure S2: Tow-Fish surface samples:** Relationship of salinity vs. dissolved (DFe) and leachable particulate Fe ($LP_{Un}Fe$) in surface waters. The Fe concentration along the y-axis is represented in a logarithmic scale. We applied a linear regression, to validate the relationship

163 between the DFe, $LP_{Un}Fe$ and salinity (not shown). With exception of the low salinity data

164 point at 33.25 psu, the DFe and $LP_{Un}Fe$ vs. salinity data achieved an $R^2$ of 0.46 and 0.38,

165 respectively.

[Figure]

168 **Figure S2S3: SAPS samples:** Relationship between leachable  (LP) and refractory

169  (RP) Fe, Mn, and Al. Due to the high proportion of RP (98.9 – 99.2% for Fe) in

170 the particulate fraction, using the particulate fraction, P, instead of RP changes the linear

171 regression with LP just very little.

[Figure]

**Figure S4: SAPS samples:** Relationship between leachable  Fe, Mn and Al.

[Figure]

**Figure S5: OTE-water sampler:** Average dissolved Fe concentration between 100 and

400 m water depth versus distance to the coast line of South Georgia in kilometre.

---

## Author Response (AR3)

We would like to thank both anonymous reviewers and the editor of BGD for their constructive comments and help to improve our manuscript. We corrected the two typos and re-submitted a new version of our manuscript and the supplementary material section. However, we are looking forward to find our manuscript soon online in Biogeosciences.

With best regards,

Christian Schlosser